# The cryo-EM structure of homotetrameric attachment glycoprotein from langya henipavirus

Yingying Guo [1,4] ✉, Songyue Wu [2,3,4], Wenting Li [2,3,4], Haonan Yang[1,4], Tianhao Shi[1], Bin Ju [2,3] ✉, Zheng Zhang [2,3] ✉ & Renhong Yan [1] ✉

Langya Henipavirus (LayV) infection is an emerging zoonotic disease that has been causing respiratory symptoms in China since 2019. For virus entry, LayV's genome encodes the fusion protein F and the attachment glycoprotein G. However, the structural and functional information regarding LayV-G remains unclear. In this study, we revealed that LayV-G cannot bind to the receptors found in other HNVs, such as ephrin B2/B3, and it shows different antigenicity from HeV-G and NiV-G. Furthermore, we determined the near full-length structure of LayV-G, which displays a distinct mushroom-shaped configuration, distinguishing it from other attachment glycoproteins of HNV. The stalk and transmembrane regions resemble the stem and root of mushroom and four downward-tilted head domains as mushroom cap potentially interact with the F protein and influence membrane fusion process. Our findings enhance the understanding of emerging HNVs that cause human diseases through zoonotic transmission and provide implication for LayV related vaccine development.

Hendra virus (HeV) and Nipah virus (NiV), which belong to the *Henipavirus* (HNV) genus within the *Paramyxoviridae* family, are highly pathogenic zoonotic paramyxoviruses known to cause severe and often fatal diseases in human[1-3]. In addition to HeV and NiV, several other HNVs have been identified in wild animals or human patients, including Mojiang virus (MojV), Cedar virus (CedV), Ghanaian bat virus (GhV), Gamak virus (GAKV), Daeryong virus (DARV), and Angavokely virus (AngV)[4]. A newly discovered HNV, called Langya Henipavirus (LayV), has recently been identified in febrile patients in eastern China and is associated with respiratory symptoms in humans[5,6]. It is believed that shrews serve as the natural reservoir for LayV, and the virus can cross over into the human population either directly or through an intermediate animal host[7]. Infected patients primarily experience symptoms such as fever, cough, nausea, headaches, and fatigue.

Although the pathogenicity and epidemiological characteristics of LayV are not yet well understood, it is suspected to have zoonotic potential[8-10]. Consequently, further research is necessary to enhance our understanding of the viral entry mechanism.

All paramyxoviruses encode two envelope proteins responsible for binding and entry of the virus into the cell. These proteins are the attachment glycoprotein (referred to as HN, H, or G protein depending on the genus, also be called "receptor binding protein, RBP") and the fusion protein (known as F protein)[11,12]. The RBP interacts with specific receptors on the host cell surface, triggering a series of conformational changes in the F protein, which enables the virus to enter the cells[13,14]. Both the F and RBP are important targets of the humoral immune response, and neutralizing antibodies against these proteins provide protection against paramyxovirus infections[15-19]. Studies have

[1]Department of Biochemistry, School of Medicine, Key University Laboratory of Metabolism and Health of Guangdong, Institute for Biological Electron Microscopy, Southern University of Science and Technology, Shenzhen 518055 Guangdong, China. [2]Institute for Hepatology, National Clinical Research Center for Infectious Disease, Shenzhen Third People's Hospital, Shenzhen, China. [3]The Second Affiliated Hospital, School of Medicine, Southern University of Science and Technology, Shenzhen, China. [4]These authors contributed equally: Yingying Guo, Songyue Wu, Wenting Li, Haonan Yang. ✉e-mail: guoyingnba@163.com; jubin2013@163.com; zhangzheng1975@aliyun.com; yanrh@sustech.edu.cn

reported that the NiV-G adopts a tetrameric type II membrane protein structure, as observed in resolved soluble tetrameric ectodomain structures[20–24]. The tetrameric structure of NiV-G protein consists of an intertwined homotetramer with an N-terminal four-helix bundle (the stalk), an interlaced β sandwich (the neck), and four β propeller head domains with two upward and two downward orientations[23]. Disulfide bonds formed between individual protein units in the neck and stalk regions contribute to the stabilization of the tetramer[23]. However, the detailed mechanisms underlying the activation of the G protein and its coordination with the F protein in HNVs remain unclear.

The analysis of LayV's genome reveals its similarity to MojV, which was initially identified in southern China in 2012[5]. LayV-G protein shares an overall sequence identity of 86% with MojV-G, and the receptor binding region exhibits significant conservation (Supplementary Fig. 1). Previous research has indicated that MojV-G does not bind to any known paramyxovirus receptor[25], while the binding capability of LayV-G to known paramyxovirus receptors remains uncertain. Furthermore, despite extensive research on the structures of HNV attachment glycoproteins, including the truncated monomeric receptor binding domain alone or in complex with receptors, as well as the tetrameric assembly of NiV-G protein, the complete structure of the full-length G protein, including the transmembrane helices in HNVs, remains elusive[23,25–30]. In this study, we conducted a comprehensive investigation into the receptor binding ability of LayV-G and its cross-reactivity with monoclonal antibodies (mAbs) that target the MojV-G and HeV-G. We also present the first near full-length cryo-EM structure of LayV-G protein, resolved at a high resolution of 2.8 Å. Our findings reports a homotetrameric architecture that distinguishes LayV-G from all known paramyxovirus RBPs structures.

## Results

### LayV-G is antigenically distinct from HeV-G or NiV-G
To characterize the molecular mechanism of LayV-G, we first purified the extracellular domain (residues 63 to 624 amino acids, short as: a.a) of LayV-G by recombinant expression. The C-terminal Flag-tagged LayV-G appears to be stable and homogeneous on size exclusion chromatography (SEC) (Fig. 1a, b), and the UV absorption peak of LayV-G was analyzed using non-reducing and reducing SDS-PAGE by coomassie brilliant blue staining and western blot, indicating the tetrameric state of LayV-G (Supplementary Fig. 2). We then sought to investigate whether LayV-G could interact with ephrinB2 and ephrinB3 in vitro, as they have been identified as the cell entry receptors of HeV and NiV[31–34]. The results showed that LayV-G couldn't bind to the well-characterized human ephrinB2 and ephrinB3 receptors after one hour incubation and SEC analysis (Fig. 1a, b). Conversely, binding experiments between NiV-G and ephrinB2 or ephrinB3 demonstrated distinct differences (Supplementary Fig. 3). Gel filtration exhibited noticeable peak shifts compared to the monomer, and reducing-SDS-PAGE analysis directly confirmed complex formation between NiV-G and ephrinB2 as well as ephrinB3. In addition, we performed binding assays of LayV-G with ephrinB2 and ephrinB3 using Biolayer interferometry (BLI, Fig. 1c) and ELISA (Fig. 1d). The results revealed significantly lower affinity compared to the positive control, NiV-G, and no detectable binding, similar to the negative control (kinetic buffer without an analyzer). Moreover, soluble LayV-G was not able to bind full-length ephrinB2 or ephrinB3 expressed on cell surface (Supplementary Fig. 4).

To assess the antigenicity of LayV-G protein in comparison to MojV-G, HeV-G and NiV-G proteins, we conducted experiments using human mAbs (HENV-117, HENV-151, HENV-165, HENV-45, HENV-242, HENV-103, HENV-78, HENV-72, and HENV-160) that recognize at least six distinct major antigenic sites (A–F) on HeV-G protein and MojV-G mAbs (6E5-LY, 10G2-LY)[35]. We performed ELISA to evaluate the binding of these mAbs to LayV-G and observed that all of them exhibited

notable binding to HeV-G and showed partial binding to NiV-G (HENV-117, HENV-45, HENV-242, HENV-103, and HENV-78), consistent with previous findings[36]. However, none of these mAbs displayed cross-reactivity with LayV-G and MojV-G, indicating that the major antigenic sites present in HeV/NiV-G are not present in LayV-G, or MojV-G (Fig. 2a). Additionally, a commercial anti-LayV-G polyclonal antibody derived from rabbits also showed minimal cross-reactivity with HeV-G and NiV-G but appreciable cross binding to MojV-G (Fig. 2b). Recombinant monoclonal antibody to MojV-G 6E5-LY but not 10G2-LY exhibited cross-reactivity with LayV-G, indicating that LayV-G and MojV-G are partially related antigens. Neither of the MojV-G mAbs bound to HeV-G or NiV-G (Fig. 2c). Taken together, these results suggest that LayV-G and MojV-G have distinct antigenicity compared to the canonical G protein of HNVs.

### The overall architecture of the LayV-G homotetramer
To systematically gain insights into the structure of LayV-G protein, we expressed and purified the full-length G protein in mammalian cells using detergent (Supplementary Fig. 5a). Employing single-particle cryo-electron microscopy (cryo-EM) technology, we successfully obtained a high-resolution 3D map of the extracellular domain of LayV-G protein at 2.8 Å resolution, revealing a distinct tetrameric architecture (Fig. 3a, Supplementary Figs. 5–7, and Supplementary Table 1). A sphere mask in Local refinement method of cryo-SPARC 3.3.1 was applied to resolve the TM domain of the full-length LayV-G protein. As a result, we obtained a low-resolution TM map with four cylindrical TM helices (Fig. 3a and Supplementary Figs. 6b and 7b).

While LayV-G protein forms a homotetramer on the virion, similar to other paramyxovirus members, its overall architecture stands out (Supplementary Fig. 8). The LayV-G homotetramer exhibits a mushroom-like structure, featuring a central four-helix bundle stalk (referred to as the 4HB domain) and a unique umbrella-like conformation formed by the four head domains (Fig. 3a, b). The umbrella-like head domains drape from the middle section of the 4HB, revealing the top of the 4HB stalk. The high-resolution structure enables to build a detailed model of LayV-G protein includes an N-terminal transmembrane region (TM) spanning residues 34 to 66 a.a, a slightly bent helix stalk (residues 67 to 142 a.a), a flexible linker (residues 166 to 186 a.a), and a downward-tilted head (residues 187 to 607 a.a) in each protomer (Fig. 3b, c). Notably, the cryo-EM map lacks density for three regions: the intracellular domain and several linker amino acids (residues 143 to 166 a.a), previously referred to as the neck domain (dashed lines in Fig. 3c, d). Moreover, an N-glycosylation site is present at amino acid Asn189 on the head domain of each protomer, which is unique to LayV (Fig. 3b–d).

### The head domains of LayV-G are nearly identical to that of MojV-G
The head domains of LayV-G (residues 187 to 607 a.a) demonstrate a globular six-bladed β propeller fold, wherein each blade consists of four antiparallel β-strands that are interconnected by seven internal disulfide bonds (Cys188-Cys604, Cys219-Cys243, Cys285-Cys298, Cys379-Cys396, Cys384-Cys502, Cys494-Cys506, and Cys568-Cys577) (Figs. 3c, d and 4a). Comparative analysis indicates a high degree of similarity between the head domain structures of LayV-G and MojV-G, with a root mean square deviation (rmsd) value of 0.706 Å. Additionally, the head domain shares certain structural characteristics with other members of the HNV genus, although there are some distinct differences when comparing the Cα atoms rmsd values (rmsd of 1.241 Å with NiV, 1.236 Å with HeV, 1.222 Å with GhV, and 1.099 Å with CedV) (Fig. 4b).

One notable difference is the presence of an additional tip structure in LayV-G and MojV-G, consisting of two antiparallel β strands from the N- and C-termini, which are further stabilized by disulfide bonds between Cys188 and Cys604 in blade5 (Fig. 4a). This

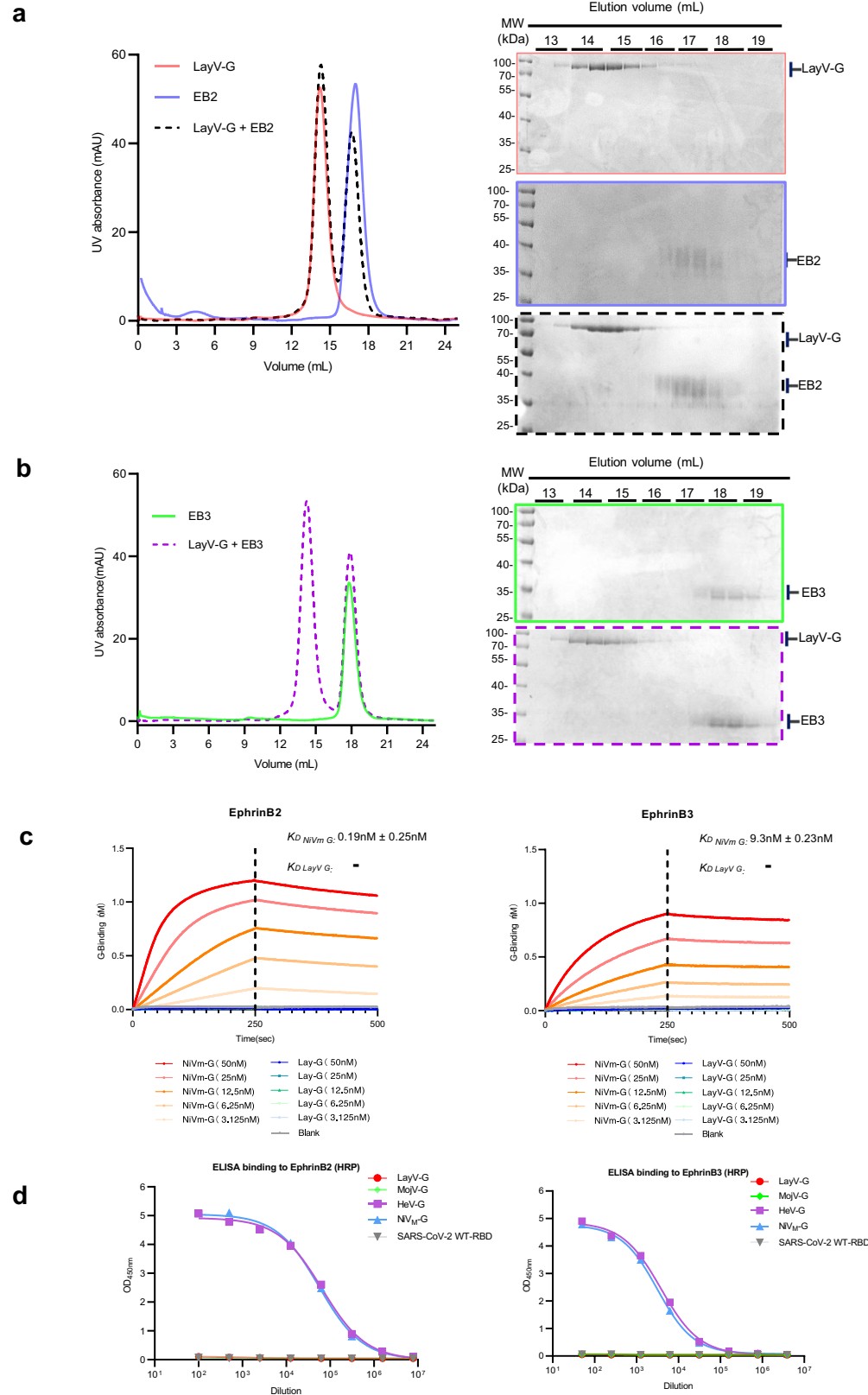

tip structure, which includes the Asn189 glycosylation site, is absent in the structures of NiV, HeV, GhV, and CedV (Fig. 4b). In the LayV-G homotetramer with a mushroom-like shape, these tip regions are positioned at the apical end of the mushroom cap (Fig. 3b). Another distinct region is observed in blade4, where two α-helices are connected by a short turn and stabilized by two disulfide bonds (Cys379-Cys396, Cys384-Cys502). This region shows similarities between

LayV-G and MojV-G, but slight differences compared to NiV, HeV, GhV, and CedV, potentially indicating a different receptor binding pattern.

Interestingly, the head domains of LayV-G adopt a "head-down" conformation, which is unexpected. Previous studies have demonstrated that the head domain of MojV-G deviates structurally from other paramyxovirus receptor-interacting surfaces and is unable to

**Fig. 1 | Biochemical characterization of the LayV-G protein and binding affinity with Henipavirus receptors ephrinB2 and ephrinB3. a, b** Comparison of gel-filtration profile of LayV-G, human ephrinB2, ephrinB3. After LayV-G co-incubation with human ephrinB2/B3; LayV-G (Red) cannot form a complex with human ephrinB2 (Blue) or ephrinB3 (green) in gel-filtration (Shown on the left panel). The reducing-SDS-PAGE of fractions collected from gel-filtration (Shown on the right panel). **c** Bio-layer Interferometry (BLI) data of binding affinity of ephrinB2 (left) & ephrinB3 (right) and purified LayV-G and NiV-G. LayV-G (Blue scatter from dark to light) and NiV-G (Orange scatter from dark to light) were used as analytes in solution, with concentrations ranging from 50 nM to 3.125 nM. **d** ELISA binding of serious diluted soluble recombinant ephrinB2 and ephrinB3 conjugated with HRP to various HNV-G ectodomain (LayV-G (red), MojV-G (green), HeV-G (purple) and NiV$_M$-G (blue)), SARS-CoV-2-WT (grey) served as a negative control. One representative curve of three independent experiments performed in technical duplicate are shown. EphrinB2-HRP and ephrinB3-HRP were separately serial diluted starting with 1:100 and 1:50. Source data are provided as a Source Data file.

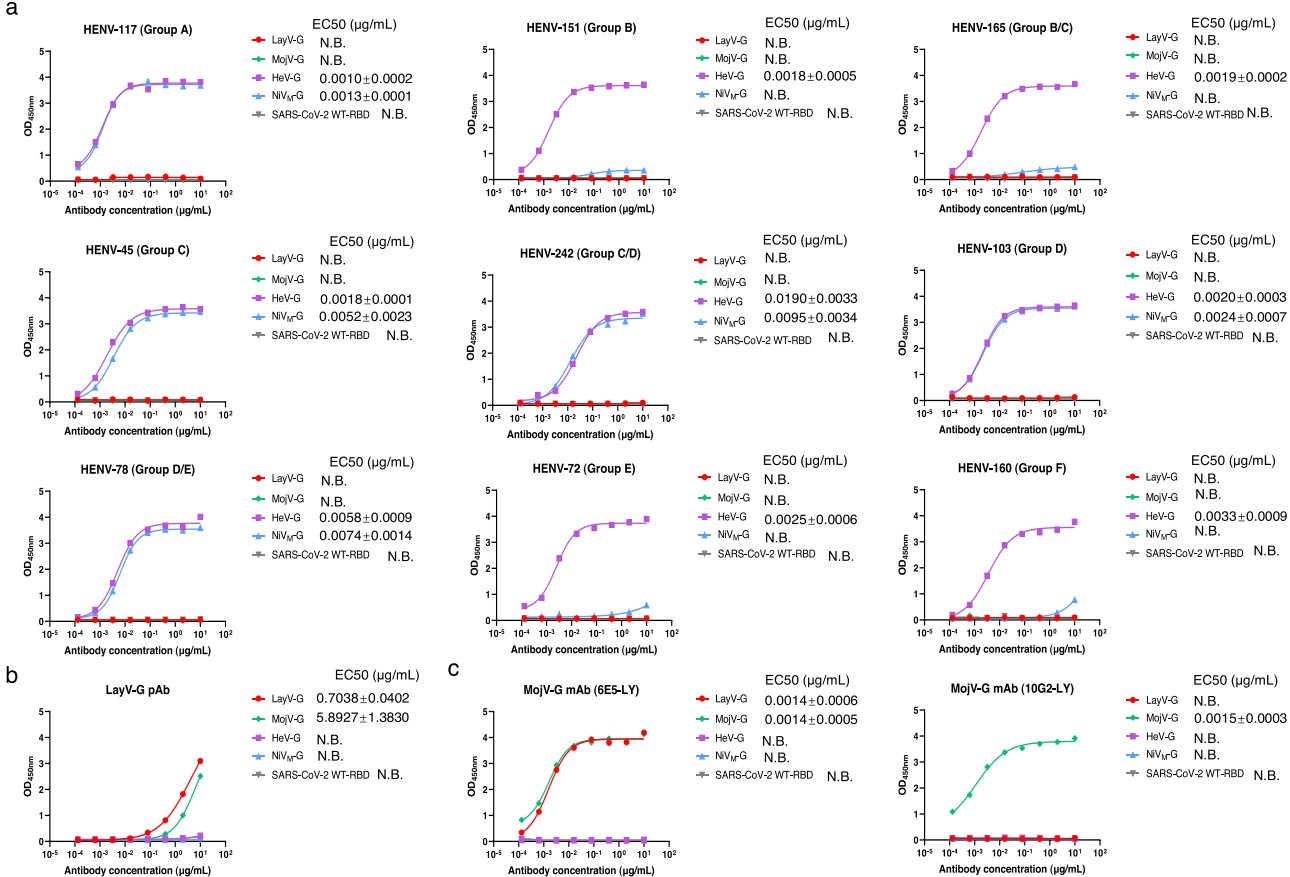

**Fig. 2 | The antigenicity of LayV-G protein in comparison to HeV-G, NiV-G and MojV-G proteins. a** Indirect ELISA detection of representative human anti-HeV-G mAbs from nine groups (A, B, B/C, C, C/D, D, D/E, E, F) binding to recombinant G proteins ectodomain of LayV-G (red), MojV-G (green), HeV-G (purple) and NiV$_M$-G (blue). SARS-CoV-2 WT-RBD (grey) served as a non-reactive antigen for antibody targeting G protein of *henipaviruses*. **b, c** ELISA detection of rabbit derived anti-LayV-G polyclonal antibody and two MojV-G mAbs against soluble LayV-G, MojV-G, HeV-G, NiV$_M$-G, and SARS-CoV-2 WT-RBD are shown. One representative curve of three independent experiments performed in technical duplicate are shown. The 50% effective concentration (EC50) values of tested antibodies are calculated on mean values of three independent experiments. Source data are provided as a Source Data file. Data are represented as mean ± SD. N.B. no binding.

interact with known paramyxovirus receptors. Consistent with this, LayV-G also fails to bind to the well-characterized human ephrinB2/B3 receptor of other HNVs (Fig. 1). These findings suggest that LayV-G and MojV-G may share a similar host-cell recognition mechanism and provide important clues to explain the inability to interact with known paramyxovirus receptors in vitro.

## The interface between head and stalk

The structure of the LayV-G tetramer reveals an internal symmetry, where two heads tilted downward are positioned in the middle of the 4HB domain and interact with adjacent helix bundle stalks from neighboring protomers (Fig. 5a). These interactions are mediated by specific amino acid residues. In head1, the residues Asp174 and Asp325 interact with Asn121 on the first helix bundle (HB1) (Fig. 5b, Supplementary Fig. 9a). In head2, the residues Asn208, Gly262, and Gln263 interact with Asn121, while Lys261, Ser269, and Leu271 form a hydrogen bond network with Lys103 (HB1), as well as Ser112 and Asn108

(HB2) (Fig. 5b, c, Supplementary Fig. 9a). Furthermore, the adjacent heads also engage in interactions at the outer edge through several polar amino acids. These interactions include Asn359 in head1 and Glu588 and Gln586 in head2 (Fig. 5d). It is noteworthy that head2-simultaneously binds to both HB1 and HB2, facilitated by the bent conformation of HB2 (Supplementary Fig. 7).

## The distinct features of 4HB stalk

The observed LayV-G stalk spans residues 34–142 a.a., including a putative transmembrane (TM) domain from residues 34 to 66 a.a (Figs. 3c and 6a). Notably, within the internal core region of the four-helix bundle (4HB), several ions' densities were observed (Fig. 6b, c). Particularly, Cys141 did not show clear formation of disulfide bonds with adjacent HB stalks (Fig. 6b). Additionally, the benzene rings of Tyr130, Phe126, and Phe89 potentially engage in cation-π interactions with cations within the internal region of 4HB (Fig. 6b, c). These ions may play a crucial role in stabilizing the tetramer, which

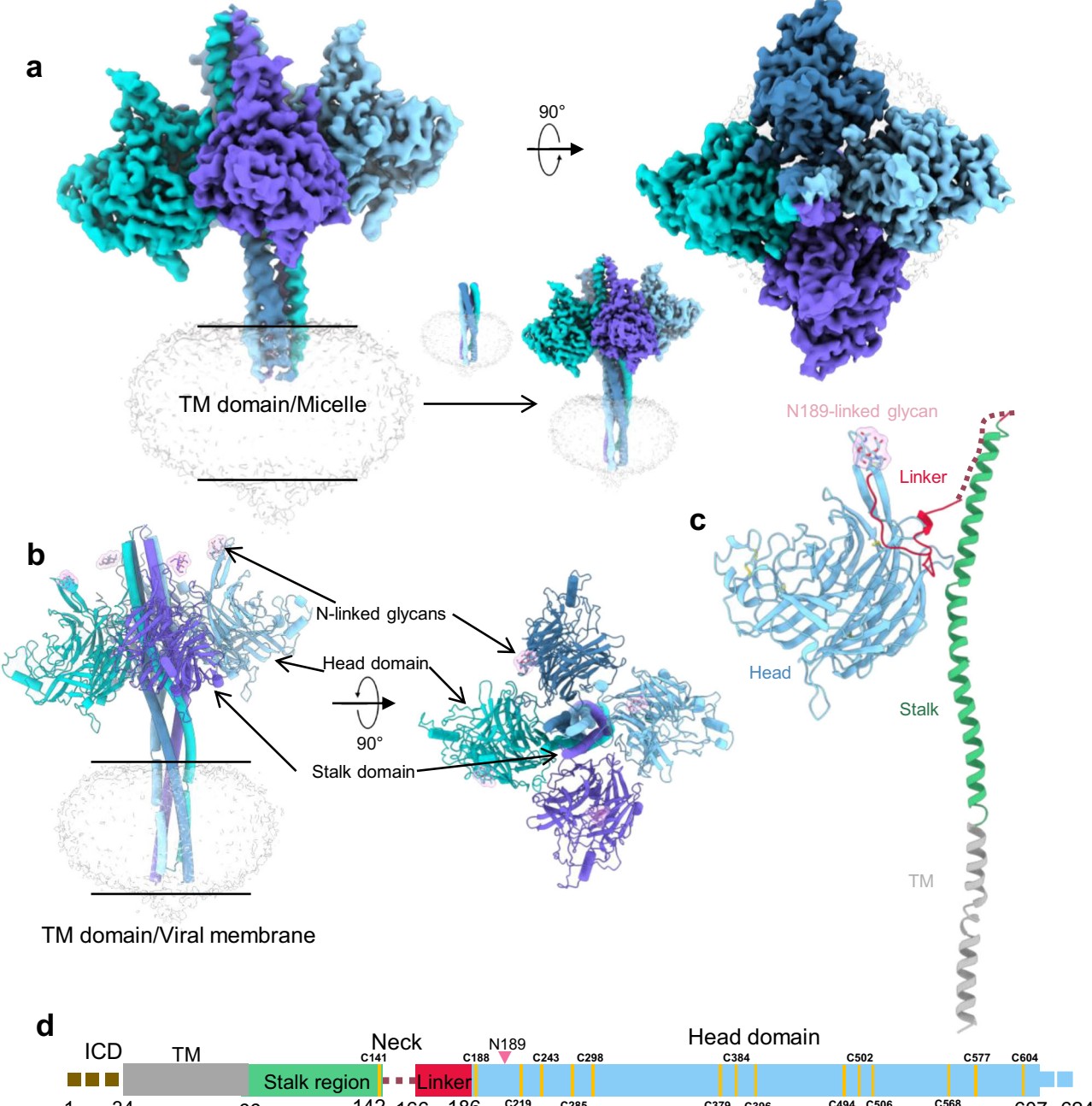

**Fig. 3 | Architecture of the nearly full-length LayV-G. a** The left panel shows the overall cryo-EM map of LayV-G (EMDB ID: EMD-36741), while the right panel displays top views of the overall structure. The structure consists of four homologous protomers, which are colored in blue, cyan, purple, and light blue, respectively. The low-resolution local refinement map of transmembrane was superimposed to the whole map. **b** The LayV-G model is represented in a cartoon diagram (PDB ID: 8JZB). The protomers are colored as described in **a**. The glycosylation site surface is highlighted in pink color. **c** In this panel, a single protomer is depicted. The head region is shown in light blue color, the glycosylation site surface in pink, the linker in red, the stalk in green, and the transmembrane (TM) domain in gray. **d** A linear representation of the full-length LayV-G is displayed. The intracellular domain (ICD) is colored in khaki, the transmembrane domain (TM) in gray. Unresolved density is indicated by a dashed line, while resolved density is represented by rectangles.

distinguishes LayV-G from other paramyxovirus attachment glycoprotein tetramer structures reported previously. Additionally, we made an intriguing observation that the surface of the 4HB exhibits a distinctive electrostatic landscape, characterized by diverse patterns of positive and negative potentials in the lower half region (Fig. 6e). This suggests the presence of strong electrostatic interactions between the 4HB domain of G and the F protein, which may play a role in triggering the conversion of F from the prefusion to the post-fusion state[36].

## Discussion

The emergence of new HNVs, including LayV and MojV in various animal species such as bats, rodents, and shrews, has raised concerns among the public and researchers[37–40]. LayV, a novel virus identified in febrile patients with a history of animal exposure in China, exhibits significant genetic similarity to MojV, another highly pathogenic HNV found in southern China[5,40]. Based on their close genetic relatedness and comparison to members of the HNV genus, both LayV and MojV could be classified as a new species within this viral group[5]. The amino

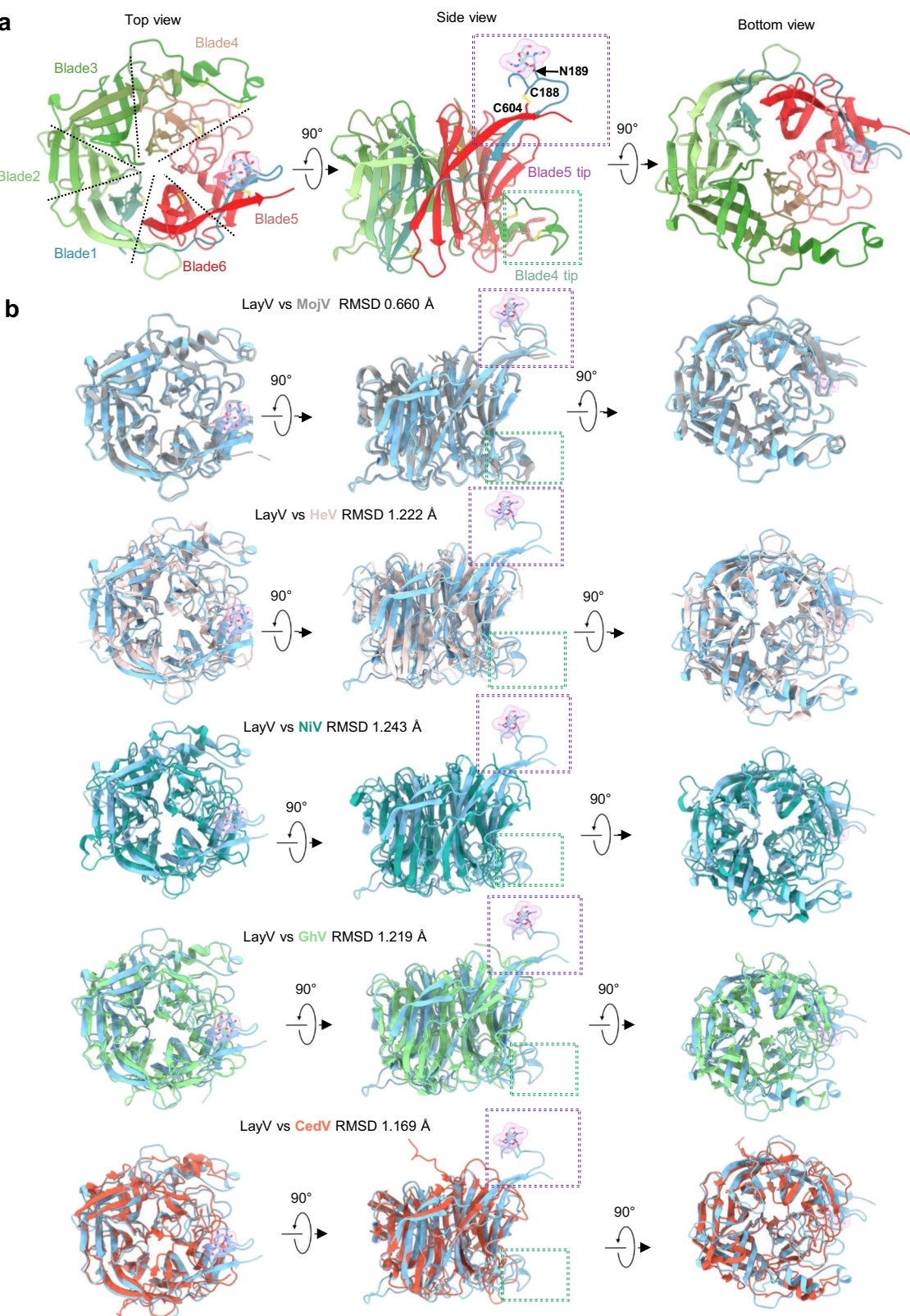

**Fig. 4 | The feature of head domain of LayV-G and structure alignment with several HNVs. a** The left panel shows the top view from a vertical direction of the viral membrane (PDB ID:8JZB). The middle panel displays the side view, while the right panel depicts the bottom view of the head domain of LayV-G. The head domain is colored in a gradient from light blue, green, to red, representing the N-terminus to C-terminus orientation. Two special tips of blade4 and blade5 were boxed by a green and a purple dotted line boxed, respectively. Cysteine residues that form disulfide bonds in two tips are shown as sticks and labeled. **b** A structure alignment is performed with five typical HNVs attachment glycoprotein. The alignment includes MojV (colored in gray, PDB ID: 5NOP), HeV (colored in light pink, PDB ID: 2VSK), NiV (colored in deep green, PDB ID: 2VWD), GhV (colored in green, PDB ID: 4UF7), and CedV (colored in orange, PDB ID: 6P7Y).The green and purple dotted line boxes exhibit the special tips at the head domain of blade4 and blade5 in LayV-G are almost identical to those in MojV, and different from HeV, NiV, GhV and CedV.

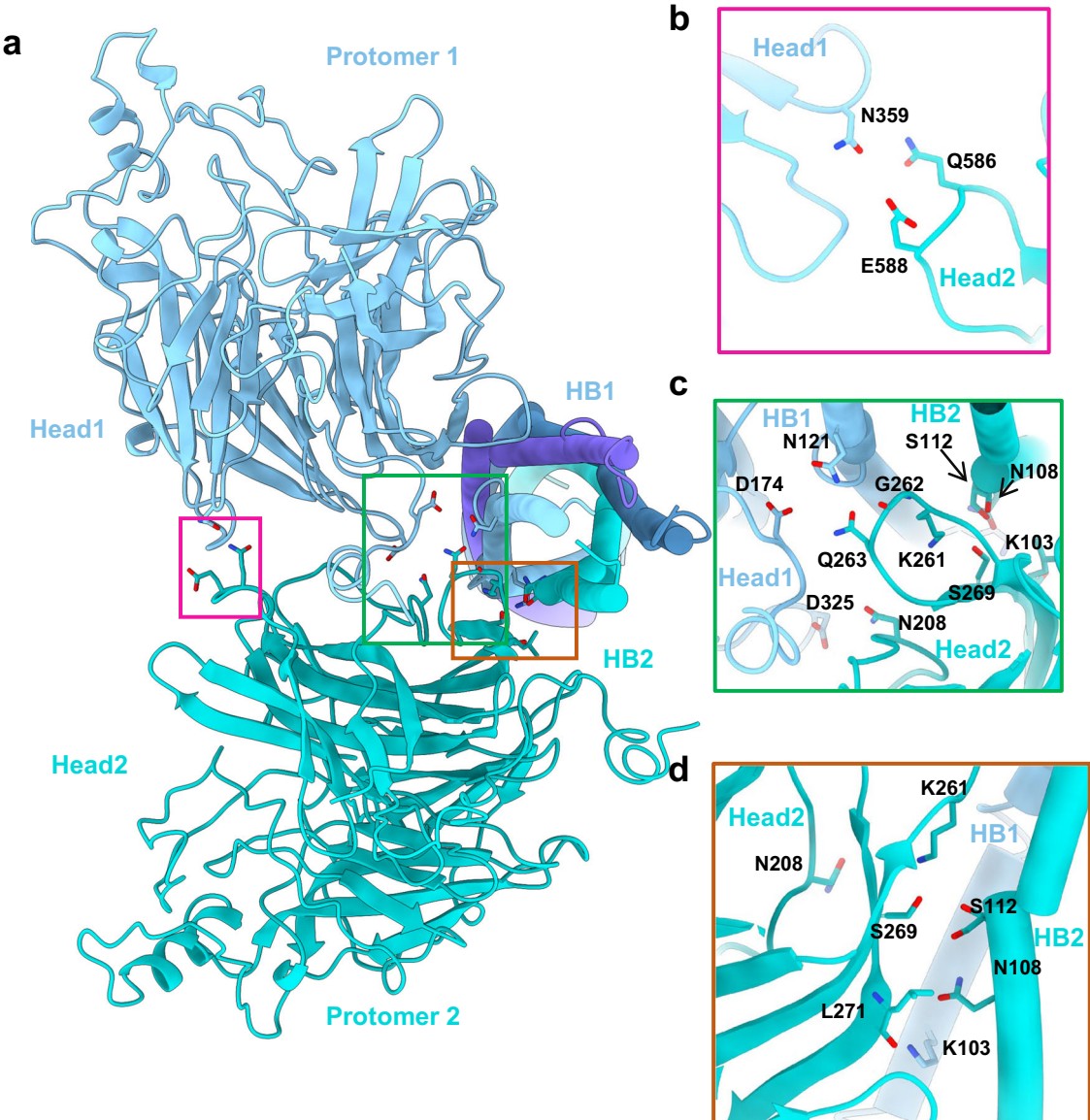

**Fig. 5 | The head and HB domain interaction. a** Two adjacent protomers (PDB ID:8JZB) are depicted, with one shown in blue color and the other in cyan color. This panel displays a detailed view of the interface between head1 from protomer1 and head2 from its adjacent protomer2. This view is from the top. **b–d** Amino acid interaction details in the interface.

acid sequence comparison of LayV-G and MojV-G reveals an 86% sequence identity and 94% sequence similarity, while LayV-G shares only 19% identity and 64% similarity with NiV-G (Supplementary Fig. 1). To date, no comprehensive reports on the full-length or ectodomain of LayV-G or MojV-G have been published, except for a crystal structure analysis of the MojV-G head[25]. Based on this prior structural and functional research, it was observed that the head domain of MojV-G displays unique antigenic characteristics and employs a distinct mechanism for host-cell recognition when compared to other members of the HNV genus[25]. Here, we investigated the antigenic divergence and receptor-binding properties of full-length LayV-G, MojV-G and canonical HNV-G proteins (HeV-G, NiV-G). We found that the several human mAbs, specifically target the HeV-G and NiV-G proteins[35], failed to bind to LayV-G and MojV-G (Fig. 2a). We also used two mAbs of MojV-G to assess antigenicity of attachment glycoprotein of HNV, and found that one of MojV-G mAb was able to bind to LayV-G antigen with the desired activity (Fig. 2b). Interestingly, these findings suggest that LayV-G exhibits unique antigenic properties, differing from the G protein found in typical HNVs. It also appears to

share a significant antigenic resemblance with MojV-G, possibly owing to similarities in both primary and quaternary structures.

In our structure, LayV-G displayed a unique homotetrameric architecture with a mushroom-like shape and four head-down conformations. As expected, when comparing the head domain alone between LayV and MojV, we observed a high degree of similarity, particularly in the putative receptor binding site (Fig. 4). LayV and MojV may share a similar architecture and receptor binding mode. Moreover, two independent research groups reported the structures of LayV-F in both the prefusion and post-fusion states, which provided crucial insights into the activation of LayV-F[36,41]. One of these studies has demonstrated that LayV-F and MojV-F exhibit distinct glycosylation modifications and antigenic profiles, despite their structural similarity to NiV[41]. According to the results reported by Ilona et al.[25], the ectodomain of MojV-G contains only four potential N-linked glycosylation sites, while the β-propeller domain of MojV-G exhibits a distinct feature of low glycolysis compared to HNV-RBPs[21,28,42–45]. In our near full-length structure of LayV-G, only a single glycosylation site (Asn189) is observed. Due to the absence of full-length MojV-G structure data, it

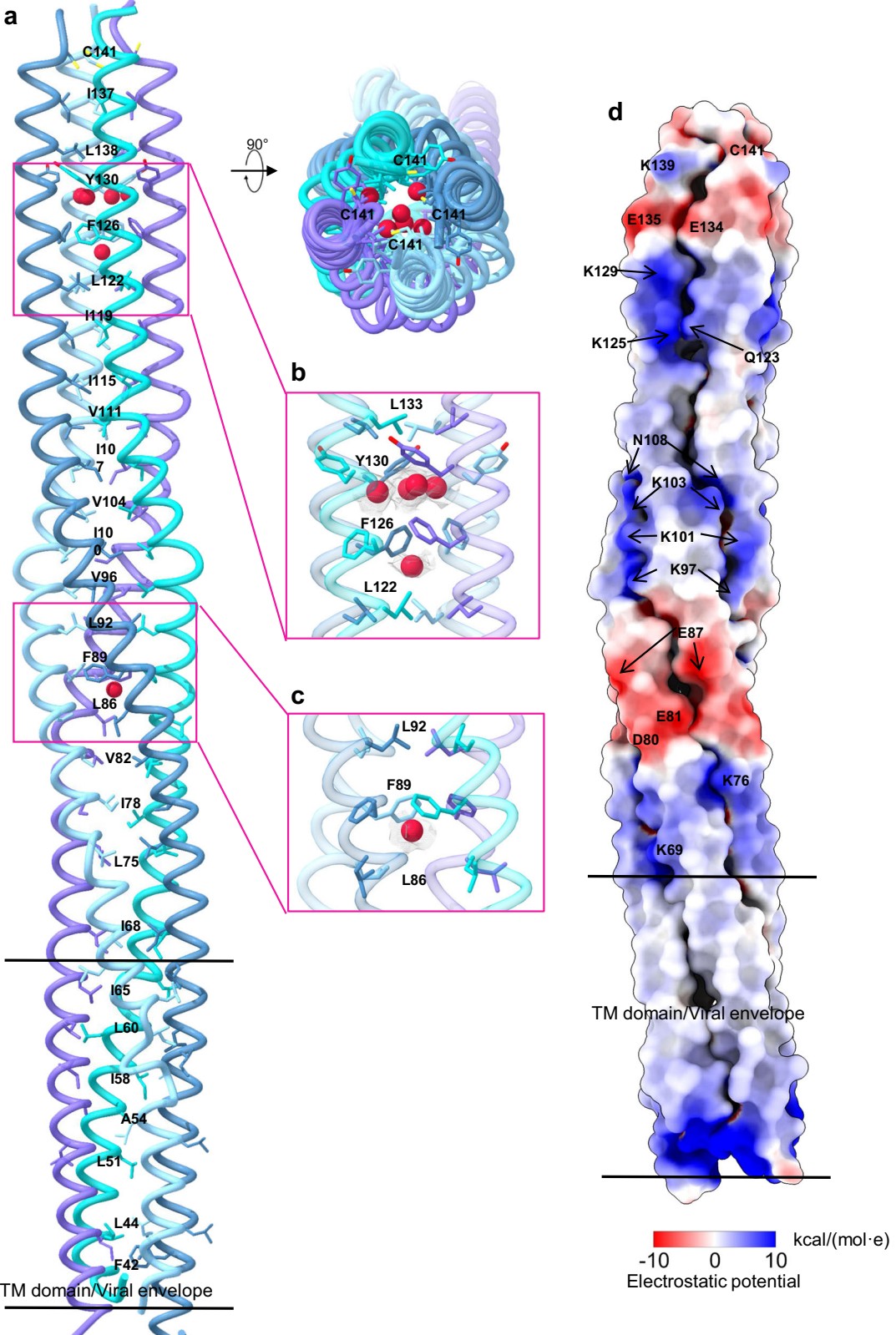

**Fig. 6 | The 4HB domain features. a** Four HB domains (PDB ID:8JZB) are represented by a ribbon diagram, with each domain colored in blue, light blue, cyan, and purple, respectively. The residues of the hydrophobic core are shown as sticks, emphasizing their location within the domains. **b**, **c** Cryo-EM density corresponding to putative sodium ions (red color) at 0.22 level in ChimeraX 1.6.1. and the surrounding amino-acid residues. **d** Electrostatic surface of 4HB domain. Electrostatic potential value was estimated in ChimeraX 1.6.1 using coulombic calculation method.

remains uncertain whether the glycosylation profiles of MojV-G are consistent with LayV-G. Therefore, a thorough investigation is required to ascertain the impact of glycosylation on the virulence and antigenic cross-reactivity of LayV-G.

The membrane fusion and entry process of most paramyxoviruses is reliant on a poorly understood activation step that involves receptor recognition, activation of the F-protein by the RBP[46,47], and subsequent membrane fusion[48,49]. In a recent study conducted by Tara et al., the resting state RBP-F complex of human parainfluenza virus 3 was visualized on the surface of authentic viruses[50]. This visualization provided insights into the mechanism by which the RBP head domains helps stabilize the pre-fusion state of the F-protein prior to receptor engagement. The stalk domain of RBP plays a crucial role in F activation and determines the specificity of the virus, while the rotation of the RBP heads may induce the movement of the stalk helices in relation to one another[47,49,51–54]. Three intriguing questions for LayV-G possible interactions with the F protein and with receptors can be raised. (i) How does the resting state of LayV-G contribute to the stabilization of the metastable prefusion state of LayV-F? (ii) If the LayV-G receptor binding site is situated on the same side as other paramyxoviruses, what factors facilitate the transition from the "down" to "up" conformation of the head domains? (iii) How does the receptor engagement trigger the metastable pre-fusion LayV-F to undergo the series of structural transitions that result in fusion of the viral and cellular? Taken together, we have proposed a hypothetical working model for the infection of LayV based on previous studies[18,22,36,50] (Fig. 7). In the resting state, the homotetrameric LayV-G forms a stable interaction with the pre-fusion state of LayV-F. Upon binding of the presumed host receptor to LayV-G, it is plausible that the head domains of the LayV-G protein undergo a significant conformational changes. Futhermore, the diverse electrostatic features exhibited on the surface of the 4HB domain seem to play a role in

initiating the transition of LayV-F from the prefusion to the pre-hairpin state and subsequently to the post-fusion state. However, additional research is necessary to provide additional evidence and support for this proposed model. Essentially, the tetrameric structure of LayV-G showcases a distinctive supramolecular architecture, setting it apart from other HNVs. These discoveries not only enhance our comprehension of the virus's cellular entry mechanism but also offer valuable perspectives for the development of structure-based vaccine candidates[55].

## Methods

### Protein expression and purification

The full-length G protein of LayV (GenBank: UUV47206.1) was cloned into the pCAG vector (Invitrogen) with an N-terminal FLAG tag. The plasmid used for cell transfection was prepared using the GoldHi EndoFree Plasmid Maxi Kit (CWBIO). The HEK293F cells (Invitrogen) were cultured at 37 °C with 5% $CO_2$ in a Multitron-Pro shaker (Infors) at a speed of 130 rpm. Once the cell density reached $2.0 \times 10^6$ cells/mL, the plasmid was transiently transfected into the cells. For transfection of one liter of cell culture, approximately 1.5 mg of the plasmid was mixed with 3 mg of polyethylenimines (PEIs) (Polysciences) in 50 mL of fresh medium. The mixture was incubated for 15 min before adding it to the cell culture. 60 h later, Cells were collected by centrifugation at $4000 \times g$ for 10 min after 60 h of transfection and resuspended in a buffer containing 25 mM HEPES (pH 7.5), 150 mM NaCl, and a mixture of three protease inhibitors: aprotinin (1.3 μg/mL, AMRESCO), pepstatin (0.7 μg/mL, AMRESCO), and leupeptin (5 μg/mL, AMRESCO).

For protein purification, the cells were incubated with 1.5% (w/v) n-dodecyl β-d-maltoside (DDM, Anatrace) at 4 °C for 2 h. Afterward, the cells were centrifuged at $18,000 \times g$ for 1 h to remove cell debris. The supernatant was loaded onto anti-FLAG M2 affinity resin (Sigma). The resin was washed with a wash buffer containing 25 mM HEPES (pH

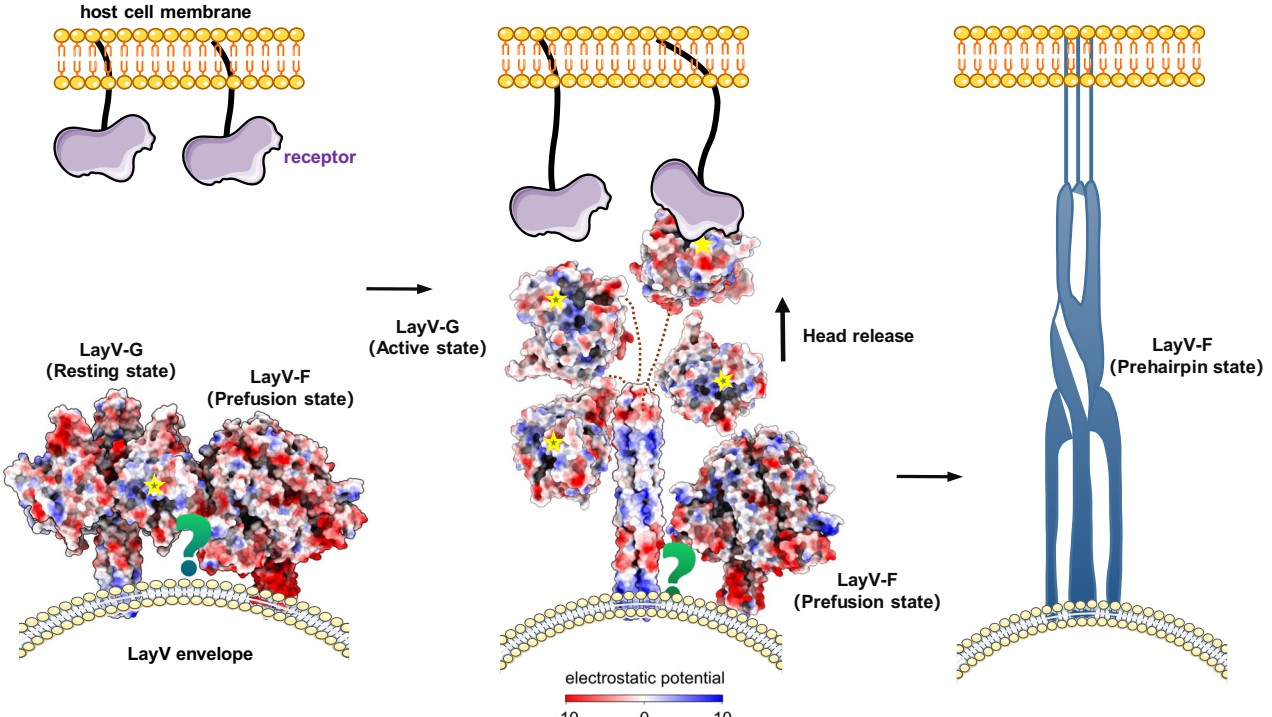

**Fig. 7 | Proposed model of LayV-G infecting host cell.** In the proposed model, the homotetrameric LayV-G (PDB:8JZB) could bind to the prefusion state of LayV-F (PDB ID: 8FEJ) in the resting state. Once the host receptor binding, the head domain of G protein might undergo a notable conformational change. Then the variegated patterns of electrostatic features on the surface of the 4HB appear to be responsible for triggering the conversion of the LayV-F from a prefusion to a pre-hairpin state and then a post-fusion state, thus triggering the membrane fusion between virus and host cells. Structure surfaces are colored by electrostatic potentials, which were estimated in ChimeraX 1.6.1 by coulombic calculation method.

7.5), 150 mM NaCl, and 0.02% GDN (w/v), followed by protein elution with the wash buffer plus 0.2 mg/mL FLAG peptide. The protein complex was then subjected to size-exclusion chromatography (Superose 6 Increase 10/300 GL, GE Healthcare) using a buffer containing 25 mM HEPES (pH 7.5), 150 mM NaCl, and 0.02% GDN. The peak fractions (14–15 mL) were collected and concentrated for EM analysis.

The extracellular domain DNA sequences of LayV-G (GenBank: UUV47206.1, amino acids: 63-624), ephrinB2 (GenBank: NP_004084.1, amino acids: 1-227), and ephrinB3 (GenBank: NP_001397.1, amino acids: 1-224) were cloned into the pCAG vector (Invitrogen) with a C-terminal 6×His tag. Approximately 1.5 mg of the plasmid was mixed with 3 mg of PEIs in 50 mL of fresh medium. The mixture was incubated for 15 min before adding it to the cell culture. 60 h later, the cell culture supernatant was harvested by centrifugation at $3500 \times g$ for 15 min at 4 °C.

To purify the LayV-G ectodomains, the supernatant was loaded onto Ni-NTA affinity resin (Sigma) twice. The resin was washed sequentially with a buffer containing 25 mM Tris (pH 8.0), 150 mM NaCl, and 30 mM imidazole. The target protein was then washed with a buffer containing 25 mM Tris (pH 8.0), 150 mM NaCl, and 300 mM imidazole in 3 column volumes to elute it. The protein was further purified by size exclusion chromatography (SEC) using Superose 6 Increase 10/300 GL (GE Healthcare) equilibrated with a buffer containing 25 mM Tris (pH 8.0) and 150 mM NaCl. The peak fractions were collected and concentrated for EM analysis. The purification steps for the ectodomains of NiV-G, ephrinB2 and ephrinB3 were the same as those for the ectodomain of LayV-G. The peak fractions were collected.

For in vitro interaction assay, the purified ECD of LayV-G, ephrinB2, and ephrinB3 were mixed in a 1:1 molar ratio and incubated on ice for 1 h respectively. The samples were then loaded onto a Superose 6 Increase 10/300 GL column that had been pre-equilibrated with a buffer containing 25 mM Tris (pH 8.0) and 150 mM NaCl. The eluted fractions corresponding to the peak were collected for further analysis using reducing-SDS-PAGE, followed by Coomassie Blue staining.

### Biolayer interferometry (BLI)
Based on biolayer interferometry using a fortéBio Octet HTC instrument. His-NTA biosensors (fortéBio, # 18-5101) were equilibrated for >600 sec in Kinetics buffer (25 mM hepes, 150 mM NaCl, 0.1% BSA and 0.02% tween 20) prior to loading with his-tagged ephrinB2 or ephrinB3 (30 μg/mL in KineticsBuffer) for 600 s. Following loading, sensors were incubated for 450 s in Blocking Buffer prior to incubation. Sensors were then incubated with analyte LayV-G (25 μg/mL in Blocking Buffer) for 250 s, then dissociated in the Kinetics buffer.

### EphrinB2/B3 binding to HNV-G ectodomain detection by ELISA
Recombinant forms of soluble LayV-G, MojV-G, HeV-G, $NiV_M$-G ectodomain (all purified in this study) and SARS-CoV-2 WT-RBD (Sino Biological, Cat# 40592-V08H) proteins were coated onto 96-well plates at 4 °C overnight. The plates were washed with PBST (containing 0.1% Tween-20 in PBS) and blocked with a solution of 5% skim milk and 2% bovine albumin in PBS at room temperature for 1 h. Serially diluted HRP (Abcam, ab102890) conjugated recombinant forms of EphrinB2 or EphrinB3 protein were added to the plates and incubated at 37 °C for 1 h. Finally, the TMB substrate (Sangon Biotech, Cat# E661007-0100) was added to each well and incubated at room temperature for 20 min. The reaction was then stopped by adding 2 M $H_2SO_4$. The readout was detected at a wavelength of 450 nm.

### Flow cytometer analysis of HNV-G protein binding to full-length ephrinB2/B3
Full-length ephrinB2 (EAX09085.1) and ephrinB3 (NP_001397.1) were separately and stably transfected into CHO cells (Cellcook Biotech Co.) using the lentivector with GFP as a marker to monitor gene expression. Briefly, pCDH-ephrinB2 or B3 and envelope vectors were co-

transfected into packaging cells. The cell culture supernatants containing the lentivirus were harvested 48 h later and further incubated with CHO cells. Afterward, stable CHO-ephrinB2 or B3 cell lines were produced via puromycin selection.

The CHO-ephrinB2 or CHO-ephrinB3 cells were stained with the LIVE/DEAD Fixable Dead Cell Stain reagent (Invitrogen, Cat# L34968) to exclude dead cells. Cells were then incubated with different HNV-G protein (20 μg/mL) at 4 °C for 30 min. Afterwards, 50 μL of diluted SureLight® APC Anti-6X His tag (Abcam, Cat# ab72579) was added to the cells and incubated at 4 °C for 30 min. Finally, cells were resuspended and binding of HNV-G protein was quantified by BD FAC-Symphony™ A3 Cell Analyzer (BD Biosciences). The amount of HNV-G on GFP positive cells was analyzed by a flowJo.

### Antibody expression and purification
Primary HeV-G monoclonal antibody (mAb) heavy chain variable (Hv) and light chain variable (Lv) domain sequences were acquired from the paper published by Doyle MP et al.[15]. Variable genes were synthesized and cloned into the expression vectors containing full-length heavy (IgG1) and light (kappa or lambda) chains by GenScript, respectively. Paired heavy and light chain plasmids were co-transfected into CHO cells to express mAbs, which were purified from the culture supernatants using protein A column (GenScript).

### Enzyme linked immunosorbent assay (ELISA)
Recombinant forms of soluble HeV-G, NiV-G, LayV-G, MojV-G (above all purified in this study) and SARS-CoV-2 WT-RBD (Sino Biological, Cat# 40592-V08H) proteins were coated onto 96-well plates at a concentration of 2 μg/mL, 100 μL/well at 4 °C overnight. The plates were washed with PBST and blocked with a solution of 5% skim milk and 2% bovine albumin in PBS at room temperature for 1 h. Primary HeV-G monoclonal antibody (mAb), MojV-G mAb (Absolute Antibody, Cat# Ab02867-10.0, Cat# Ab02868-10.0) and the anti-LayV-G polyclonal antibody (pAb) (AntibodySystem, Cat# PVV18301) were serially five-fold diluted from 10 μg/mL and applied to the wells. Plates were incubated at 37 °C for 1 h. After washing the plates 5 times with PBST, horseradish peroxidase-conjugated secondary antibodies (goat anti-rabbit IgG (H + L), or goat anti-human IgG (H + L) from ZSGB-BIO, Cat# ZB-2301, Cat# ZB-2304) were added to each well at a dilution of 1:5,000 in the blocking solution. Following a 1-h incubation at 37 °C, the plates were washed 5 times with PBST. Finally, the TMB substrate (Sangon Biotech, Cat# E661007-0100) was added to each well and incubated at room temperature for 20 min. The reaction was then stopped by adding 2 M $H_2SO_4$. The readout was detected at a wavelength of 450 nm. The 50% effective concentration (EC50) values of tested mAbs were calculated using GraphPad Prism 8.0 software by log (agonist) vs. response – Variable slope (four parameters) model.

### Western blot (WB)
The samples were prepared using 5×SDS-PAGE Loading Buffer containing 0.25 M Tris-HCl (pH 6.8); 10% SDS; 0.5% BPB; 50% glycerol; 5% (W/V), with or without 100 mM dichlorodiphenyltrichloroethane (DTT) depending on reduced or nonreduced conditions analyses. Then, load each sample into the respective lanes of the SDS-PAGE gel. Run the gel at 250 V for 45 min. After electrophoresis, place the filter paper, PVDF membrane and gel in the blot module, and insert the module into electrophoresis chamber, transferring proteins from the gel to the PVDF membrane for 90 min at 300 mA. Incubate the membrane respectively in PBST containing 5% (w/v) BSA for 2 h at room temperature, and primary antibody anti His-tag mouse monoclonal antibody (CWBIO, Cat#CW0826) at 1: 1000 diluted in PBST overnight at 4 °C, secondary antibody HRP conjugated goat anti-rabbit IgG, (CWBIO, Cat#CW0102S) at 1: 1000 diluted in PBST for 1 h at room temperature. Finally, incubate the membrane in ECL reagent for 1 min

in the dark, and then place the blot on a piece of plastic wrap and image.

## Cryo-EM sample preparation and data acquisition

The purified full-length LayV-G was concentrated to approximately 5 mg/mL before being applied to glow-discharged holey carbon grids (Quantifoil Au R1.2/1.3). Aliquots of the protein sample (3.3 μL) were placed on the grids. The grids were then blotted for 3 s or 3.5 s and flash-frozen in liquid ethane cooled by liquid nitrogen using the Vitrobot (Mark IV, Thermo Fisher Scientific). The cryo grids were transferred to a Titan Krios microscope operating at 300 kV, equipped with a Gatan K3 Summit detector and GIF Quantum energy filter. Movie stacks were automatically collected using EPU software (Thermo Fisher Scientific) with a slit width of 20 eV on the energy filter. The defocus range was set from −1.4 μm to −1.8 μm in super-resolution mode at a nominal magnification of 81,000×. Each stack was exposed for 2.99 s, with an exposure time of 0.09 s per frame, resulting in a total of 32 frames per stack. The total dose rate for each stack was approximately 50 e−/Å$^2$. Subsequently, the stacks were motion corrected using MotionCor 2.1.1.0[56]. After motion correction, the movie stacks were binned 2-fold, resulting in a pixel size of 1.095 Å/pixel. Dose weighting was then applied to the data[57]. The defocus values were estimated with Gctf[58].

## Data processing

The Cryo-EM structures of LayV-G were solved in cryoSPARC 3.3.1. Particles were automatically picked using cryoSPARC 3.3.1[59–62]. After 2D classification, the micrographs with good particles were selected, and these particles were subjected to several cycles of 2D classification, Ab-Initio Reconstruction, and multiple cycles of heterogeneous refinement without symmetry using cryoSPARC 3.3.1[63]. The good particles were selected and subjected to Local CTF Refinement with C1 symmetry, Non-uniform Refinement, resulting in the 3D reconstruction for the whole structures. The resolution was estimated with the gold-standard Fourier shell correlation 0.143 criterion[64] with high-resolution noise substitution[65]. Refer to Supplementary Fig. S6 and Supplementary Table S1 for details of data collection and processing.

## Model building and structure refinement

Predicted models of LayV-G was first obtained using Alphafold2[66], which was further manually adjusted based on the cryo-EM map with Coot 0.8.2[67]. Each residue was manually checked with the chemical properties taken into consideration during model building. Several segments, whose corresponding densities were invisible, were not modeled. Structural refinement was performed in Phenix 1.11.1[68] with secondary structure and geometry restraints to prevent overfitting.

To monitor the potential for overfitting, the model was refined against one of the two independent half maps obtained from the gold-standard 3D refinement approach. Subsequently, the refined model was validated by testing it against the other map. Detailed statistics related to data collection, 3D reconstruction, and model building can be found in the Supplementary information, specifically in Supplementary Table 1.

## Reporting summary

Further information on research design is available in the Nature Portfolio Reporting Summary linked to this article.

## Data availability

The Atomic coordinates and cryo-EM density maps of LayV-G protein generated in this study have been deposited in the Protein Data Bank (PDB) under accession codes 8JZB (LayV-G) and the Electron Microscopy Data Bank (EMDB) under accession codes EMD-36741 (LayV-G). Correspondence and requests for materials should be addressed to

R.Y. (yanrh@sustech.edu.cn). Source data are provided with this paper.

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

## Acknowledgements

We thank the Cryo-EM Facility of Southern University of Science and Technology (SUSTech) for providing the facility support. We thank Shuman Xu, Xiaomin Ma, and Lei Zhang at the Cryo-EM Center of SUS-Tech for technical support in electron microscopy data acquisition. We thank Zhenyuan Liu for technical support on computing environment. We thank Dr. Wenjing Liu for the initial cloning and protein purification. This work was funded by the National Natural Science Foundation of China (82202517 to R.Y., 82171752 to B.J.), and the National Science Fund for Distinguished Young Scholars (82025022 to Z.Z.), the Major Talent Recruitment Program of Guangdong Province (2021QNO2Y167 to R.Y.), and the the Science, Technology and Innovation Commission of Shenzhen Municipality (JSGG20220226085550001 to R.Y.).

## Author contributions

R.Y., Z.Z. and B.J. conceived the project. R.Y. and Y.G. designed the experiments. S.W. and H.Y. did the molecular cloning, protein purification. Y.G. and T.S. did the cryo-EM data collection and processing, and model building. W.L. performed the ELISA and flow cytometry experiment. All authors contributed to data analysis. R.Y. and Y.G. wrote the manuscript. Z.Z. and B.J. participated in the revision of the manuscript.

## Competing interests

The authors declare no competing interests.
