## [Peer Review File · Nature Communications]

REVIEWER COMMENTS

Reviewer #1 (Remarks to the Author):

In the present study Guo, Wu, Li, and Yang report on the biochemical and structural characterization of the Langya virus (LayV) attachment protein, G. The authors recombinantly expressed and purified LayV G and showed, *in vitro*, that, unlike other G proteins from the Henipavirus (HNV) genus, LayV G does not interact with the cellular receptors ephrin B2 or B3. They also showed that LayV G is antigenically distinct from the other members of the genus, as monoclonal antibodies against other henipaviral Gs do not cross-react with LayV G. Finally, the authors determined a cryo-EM structure of the full-length LayV G which revealed that LayV G forms a 4-fold symmetric tetramer. This is contrast to previous studies of other HNV Gs that for dimer-of-dimer type of tetrameric structures.

While the reported structure does shed insight into the differences in quaternary structure of LayV G and other HNV Gs, I feel that the entire study is rushed and many experiments as well as their analysis lack rigor.

Major concerns:

Experimental:

1 – In Figure 1, the authors use gel-filtration with pre-mixed ephrin B2 and LayV G to argue that there is no interaction between the two. Just because two proteins do not co-elute on a gel-filtration, it does not necessarily mean that there is no interactions, as low affinity or reactions with slow kinetics may not form complexes on gel-filtration. A biophysical method like SPR, BLI, ITC etc. should be used to rule out these phenomena.

2 – Also Figure 1, the SDS-PAGE gel results are shown, but there is no molecular weight marker shown, nor is it mentioned which one was used. Was this gel run under reducing or non-reducing conditions? Ephrin B2 bands on the gel are very faint and this gel is close to uninterpretable. I understand that glycosylations may affect the migration on the gel and make the bands “smeary”, but more material could have been used and/or the gel could have been run under non-reducing conditions etc.

3 – Still in Figure 1, endpoint ELISA optical density values are presented for the different antibodies. This is not a quantifiable measurement. A titration series should have been performed, and area under the curve, or EC50 values plotted.

4 -The cryo-EM structure processing seems incomplete, as per Supplementary Figure 4. No CTF refinement was performed, only one round of non-uniform refinement, presumably.

5 – The authors mention that they could not resolve the transmembrane parts of the structure, but have they attempted to do so? Focused classification and/or refinement could have been done on the micelle part to resolve the TM helices.

Figures:

1 – What are the special characters in the consensus line of Supplementary Figure 1? They are not defined in the legend.

2 - Supplementary Figure 3, the legend lacks most of the relevant information that it should contain. What is “purification by detergent”?

3 – Line 184, Supplementary Figure 6c is called out but there is no Supplementary Figure 6c. Also, the figure legend makes little sense: “head binding cause two places of stalk bend”

Minor:

1 – The title – what is so “special” about this particular protein structure?

2 – The text could use major editing to improve syntax and grammar.

Reviewer #2 (Remarks to the Author):

Guo et al. determined the first cryo-electron (cryo-EM) structure of tetrameric Langya Henipavirus (LayV) attachment glycoprotein G. To date, neither the ectodomain nor the full-length LayV-G architectures have been described; therefore, the presented structures can have high importance for vaccine and therapeutic development. Here authors pointed out architectural and antigenic differences between LayV-G and other henipaviruses. Using single particle cryo-EM, the authors highlighted that the LayV-G head domain consists of a globular six-bladed β -propeller fold and exhibited profound architectural similarity to the Mojiang virus (MojV) G glycoprotein, including an Asn189 glycosylation site. Further, the LayV-G ectodomain structure resembled a mushroom shape, with the four heads down, as opposed to other paramyxoviral proteins with two heads up and two heads down. The authors indicated that LayV-G behaves like MojV, which has been shown not to bind ephrinB2/B3, the well-characterized Nipah and Hendra cellular receptors, highlighting the need for a more comprehensive investigation of LayV and MojV receptor utilization. Here are major and minor concerns regarding this structural study.

Major comments:

1. In Figure 1a for biochemical characterization of the LayV-G protein, authors did not provide experimental controls for ephrinB2 binding either in the gel-filtration profile nor the SDS-PAGE analyses. Data in Figure 1 did not provide any further biochemical evidence that purified recombinant protein produced from LayV-G sequences is folded correctly or is a functional protein. To address this concern, including NiV-G and MojV-G binding profiles with ephrinB2/B3 as positive and negative controls in Figure 1 or supplementary Figure 2 could strengthen these data.

2. It is not clear that the purified ephrinB2 or ephrinB3 used are stable or functional proteins that resemble functional surface expressed ephrin's. Since the gel-filtration approach does not address this concern, consider using ELISA or flow cytometry to determine LayV-G binding capacity to susceptible cells, or a functional cell-cell fusion assay.

3. The authors did not indicate in the legend of Figure 1 nor the Methods section how they prepared the samples for SDS-PAGE, and which conditions, e.g. reducing or non-reducing, they used. In addition, information regarding the method of choice for protein staining must be included.

4. From the results of the SDS-PAGE image in Figure 1, it is unclear what type of putative monomeric, dimeric, or tetrameric protein species were collected from gel-filtration. The authors should include results from a Coomassie stain or results of Western Blot analysis under non-reducing conditions stained against expressed proteins.

5. The reported observation that LayV-G adopts a "heads-down" conformation is very interesting. Nevertheless, the authors should have explained or elaborated in the Discussion how this heads-down conformation compares to MojV and is related to possible interactions with the fusion protein and with receptors.

6. The authors confirmed that LayV-G does not bind ephrinB2 or ephrinB3. However, this finding had already been shown and that original reference (Zhang et al., NEJM 2022) should be cited. The following 3 primary references for important papers reporting ephrinB2 and ephrinB3 as HeV and NiV receptors are also missing and should be cited:

a) Negrete OA, Levroney EL, Aguilar HC, Bertolotti-Ciarlet A, Nazarian R, Tajyar S, Lee B. EphrinB2 is the entry receptor for Nipah virus, an emergent deadly paramyxovirus. *Nature*. 2005 Jul 21;436(7049):401-5. doi: 10.1038/nature03838. Epub 2005 Jul 6. PMID: 16007075.

b) Bonaparte MI, Dimitrov AS, Bossart KN, Crameri G, Mungall BA, Bishop KA, Choudhry V, Dimitrov DS, Wang LF, Eaton BT, Broder CC. Ephrin-B2 ligand is a functional receptor for Hendra virus and Nipah virus.

Proc Natl Acad Sci U S A. 2005 Jul 26;102(30):10652-7. doi: 10.1073/pnas.0504887102. Epub 2005 Jul 5. PMID: 15998730; PMCID: PMC1169237.

c) Negrete OA, Wolf MC, Aguilar HC, Enterlein S, Wang W, Mühlberger E, Su SV, Bertolotti-Ciarlet A, Flick R, Lee B. Two key residues in ephrinB3 are critical for its use as an alternative receptor for Nipah virus. PLoS Pathog. 2006 Feb;2(2):e7. doi: 10.1371/journal.ppat.0020007. Epub 2006 Feb 10. PMID: 16477309; PMCID: PMC1361355.

Minor comments:

1. In the Abstract, the authors claimed that they determined the full-length structure of LayV-G, although the cryo-EM map lacked densities for the intracellular domain, portion of the transmembrane domain, and the neck domain. In this case, it is more accurate to report the determination of the structure of the LayV-G ectodomain.

2. In the Abstract incomplete sentence. "In this study, we conducted experiments to investigate LayV-G's binding capabilities." Binding capabilities to what? It would be clearer if they mention actual binding to the ephrin receptor in this sentence.

3. In Figure 1b the authors did not indicate the source or company name of anti-LayV-G polyclonal antibodies used for their ELISA analysis.

4. Including MojV-G glycoprotein for screening with anti-HeV-G mAb can help to obtain a full assessment of LayV-G antigenicity. Consider using anti-MojV monoclonal or polyclonal antibodies to identify potential cross-reactivity with MojV-G to elaborate on the similarity between LayV-G and MojV-G.

5. In the Methods section Protein expression and purification, correct 2.0×10^6 cells/mL to 2.0×10^8 cells/mL.

6. In the Methods section Cryo-EM sample preparation and data acquisition, add space after the period see "...GIF Quantum energy filter. Movie stacks..."

7. In the Methods section (Model building and structure refinement), correct alphafold2 to AlphaFold 2.

8. Line 177 remove space. Correct “in head 2” to “in head2”.

9. The claimed novel structure shared between LayV-G and MojV-G, but absent in NiV and HeV, as described in paragraph at lines 151-159 “two antiparallel β strands from the N- and C-termini, which are further stabilized by disulfide bonds between Cys188 and Cys604 in blade 6 (Fig 3a)” should be better depicted and labeled in Fig. 3a so that any reader can easily identify this novel structure.

10. Acknowledgments and author contributions section: It needs to be clarified why Dr. Wenjing Liu is not a coauthor on the paper. He performed initial cloning and protein purifications, as S.W. and H.Y. Could the authors justify why the contributions of S.W. and H.Y. but not Wenjing Liu are worth authorship on the paper?

11. Line 289: Reference paper #7 Zhao, Y et al. is inappropriate. This publication describes a large-scale integrative analysis of psychiatric disorders and is irrelevant to the topic cited in the Introduction section (Line 51).

12. Line 358 and Line 371: In the References section, May A.J at el. J Virol (2023) was referenced twice as reference #34 and reference #40.

13. References for receptor-induced conformational changes in other henipaviral G proteins such as NiV or HeV should be added in the Discussion. Examples are Liu et al, 2013 and Liu et al., 2015.

Reviewer #3 (Remarks to the Author):

The manuscript by Guo et al. describes the unique characteristics of LayV’s attachment protein including a receptor different from other HNVs (ephrin B2/B3), lack of cross-reactivity with known HeV neutralizing antibodies and its unique structure with 4 downward tilted head domains (homotetrameric architecture). The manuscript could be made stronger with additional structural overlays, modeling and sequence alignment figures to clearly illustrate these differences.

Overall, with additional figures to support author’s statements and work on/revision of the discussion section, the manuscript provides valuable insight for structure-based vaccine design, the variations that

occur within the same viral family and the limitations of generalizable solutions for pandemic preparedness.

Major Comments

- 1) In figure 1 when describing the lack of cross-reactivity with known HeV antibodies, it would be helpful to add a figure that compares sequence alignments between LayV, MojV, HeV and NiV. Are there regions of conservation that could represent potential cross-reactive epitopes? Are the antigenic epitopes mapped for the antibodies described? Maybe consider structural conservation maps (i.e. consurf) and highlight antigenic epitopes for the HeV Abs tested. This sort of figure would provide stronger support the final statement (line 113-114) in the paragraph. Was binding to MojV G evaluated?
- 2) In Figure 3a, were there supposed to be zoom in figures (light grey boxes). If not, it would be helpful. In a supplementary figure, add overlays highlighting blade 6 and blade 4 between LayV/MojV and LayV/HeV, NiV or CedV.
- 3) Line 194-196, please show through overlays or side by side images, the differences between LayV G and other HNVs.
- 4) Line 203-205 – There is no strong evidence that this is the case. This also goes to the statement in the abstract mentioning “the F protein influencing membrane fusion by obstructing the charged regions of the stalk”. I would exclude “by obstructing the charged regions of the stalk”. This is a hypothesis which can be discussed in the discussion but no firm conclusions can be made at this time.
- 5) Line 233-241 – the model proposed in Figure 6 is the same as that proposed in May et al 2023. The difference is in highlighting the possible role of the electrostatic 4HB domain. I would reference May et al. when describing the proposed model.

Minor Comments

- 1) A better description of the HNV attachment protein, in its many forms (HN/H/G) may be receptor binding protein (RBP).
- 2) Line 64, need a space at end of sentence.
- 3) Line 111, anti-LayV-G polyclonal antibodies should be antibody (it is a single sample).
- 4) Line 136-7, add that the N-glycosylation site at Asn 189 is unique to LayV. Was this commented on in the discussion section?
- 5) Figure 1a – was there a positive control demonstrating binding to ephrin B2 with either NiV G or HeV G?
- 6) In figure 2 – the use of lighter shades, makes it easier to see the cryo-EM structure.
- 7) In figure 3b – use of lighter shades would make it easier to see structural divergence and/or overlap.
- 8) Please orient figure 3a and b (top) in the same direction.

- 9) What are blades? I didn't see a clear definition.
- 10) In Figure 4a, label the red boxes in the corner to correspond to b,c,d as appropriate. Again, lighter shades, especially in b, c and d would make it easier to see what the authors are trying to show.
- 11) Figure 5 legend, d electrostatic surface...should be e electrostatic surface...
- 12) In Figure 5, mark positive (blue) and negative (red) charge.
- 13) Lines 162-169 – paramyxoviral should be paramyxovirus (3 instances)
- 14) Line 175, need a space between head1.
- 15) Reference 34 and 39 are the same.
- 16) Supplementary figure 1 legend Lay-G should be LayV-G (correct twice), also consider selected different colors than red and green for those with color blindness.
- 17) Supplementary Figure 2 legend, *in vitro* should be italicized.
- 18) Supplementary Figure 3 legend, full-length should be full-length.
- 19) Supplementary Figure 3 – please include a bar of length (nm) for the EM micrographs.
- 20) Supplementary Figure 6 legend – *in vitro* should be italicized, headings a and b need to be switched.
- 21) The HeV antibodies used should be described in the methods and reporting summary.

Response to Reviewers:

We appreciate the critical comments from all three reviewers. Below please find our point-to-point responses.

Reviewer #1:

We greatly appreciate the reviewer's recognition of the significance of our study. He or she made a number of very constructive comments. We appreciate this reviewer's input and revised the manuscript accordingly.

Major concerns:

1 – In Figure 1, the authors use gel-filtration with pre-mixed ephrin B2 and LayV G to argue that there is no interaction between the two. Just because two proteins do not co-elute on a gel-filtration, it does not necessarily mean that there is no interactions, as low affinity or reactions with slow kinetics may not form complexes on gel-filtration. A biophysical method like SPR, BLI, ITC etc. should be used to rule out these phenomena.

We thank this reviewer for this critical comment. In response, we have introduced a positive control into our study: it is well-documented that NiV-G and ephrin-B2/B3 can form a stable complex. To validate this, we conducted an incubation of NiV-G and ephrin-B2/B3, followed by passage through gel-filtration. The results exhibited a significant shift in the peak position of the protein, and subsequent analysis via SDS-PAGE under reducing conditions confirmed the formation of a relatively stable complex between NiV-G and ephrin B2/B3 (Revised Supplementary Fig. 3). It is in stark contrast to the results observed for LayV-G (Fig. 1a,b), further establishing the reliability of this method as a control. Moreover, we delved deeper into the kinetics of the binding interactions between LayV-G and NiV-G with ephrin B2/B3 (as positive control) by BLI analysis. This additional investigation provides further insights into the affinity between LayV-G and ephrin B2/B3 comparing to the positive control (NiV-G) and negative control (kinetic buffer without analyser) reinforcing the notion that LayV-G could not exploit ephrin B2/B3 as host receptors (Revised Fig.1c). Besides, we also performed ELISA binding assay to determine the interaction for LayV-G and several other HNV-G (HeV-G, NiVM-G and MojV-G) with ephrin B2/B3 (Revised Fig. 1d).

Revised Supplementary Fig. 3 Biochemical characterization of the NiV-G binding to ephrin B2 and ephrin B3

Revised Fig.1c Bio-layer Interferometry (BLI) data of Binding affinity of ephrinB2

2 – Also Figure 1, the SDS-PAGE gel results are shown, but there is no molecular weight marker shown, nor is it mentioned which one was used. Was this gel run under reducing or non-reducing conditions? ephrin B2 bands on the gel a very faint and this gel is close to uninterpretable. I understand that glycosylations may affect the migration on the gel and make the bands “smeary”, but more material could have been used and/or the gel could have been run under non-reducing conditions etc.

We appreciate the valuable insight provided by the reviewer. We have included molecular weight markers and provided detailed SDS-PAGE conditions in the legends of both Fig.1a and Revised Supplementary Fig. 2a, enhancing the clarity of our methodology. Regarding the observed "smeary" appearance of ephrin B2 in our experiments, we would like to clarify that this phenomenon is attributed to the glycosylation modifications naturally present in ephrin B2, as you rightly pointed out. We conducted SDS-PAGE under non-reducing conditions to investigate this issue further (Revised Supplementary Fig. 2a). While the appearance of ephrin B2 did not change significantly under these conditions, we recognize the importance of transparency in our reporting. To reinforce the authenticity of the ephrin B2 monomer we expressed, we performed Western blot (WB) analysis (Revised Supplementary Fig. 2b). However, our binding experiments with NiV-G conducted through BLI (Revised Fig. 1c) further substantiate the natural conformation and functionality of the EB2 monomer, and the gel-filtration result of NiV G binding to ephrin B2/B3 has proved it (Revised Supplementary Fig. 3)

Revised Supplementary Fig. 2 Biochemical characterization of the LayV-G and ephrin B2 with SDS-PAGE and Western blot

Revised Fig.1c Bio-layer Interferometry (BLI) data of Binding affinity of ephrin B2

Revised Supplementary Fig. 3 Biochemical characterization of the NiV-G binding to ephrin B2 and ephrin B3

3— Still in Figure 1, endpoint ELISA optical density values are presented for the different antibodies. This is not a quantifiable measurement. A titration series should have been performed, and area under the curve, or EC50 values plotted.

We sincerely appreciate the insightful comment provided by the reviewer. We have performed additional ELISA assay with titration series of different HeV-G monoclonal antibodies and polyclonal antibody. EC50 values have been added and shown in the plots. The plots are showed in Revised Fig.2.

Revised Fig.2 The antigenicity of LayV-G protein in comparison to HeV-G, NiV-G and MojV-G proteins.

4 -The cryo-EM structure processing seems incomplete, as per Supplementary Figure 4. No CTF refinement was performed, only one round of non-uniform refinement, presumably.

We appreciate the valuable insight provided by the reviewer. To further optimize the cryo-EM computational workflow (Heterogeneous Refinement, Non-uniform Refinement, Local CTF Refinement and Local Refinement), we have improved the resolution of the LayV-G protein to 2.77 Å (Supplementary Fig. 6a). We also added a sphere mask for resolved the transmembrane parts of G protein and got a low-resolution (7Å) map, which show the trend of transmembrane region helices (Supplementary Fig. 6b).

5 – The authors mention that they could not resolve the transmembrane parts of the structure, but have they attempted to do so? Focused classification and/or refinement could have been done on the micelle part to resolve the TM helices.

We thank this reviewer for this good suggestion. We first improved the cryo-EM structure

processing and got a new density map (Revised Supplementary Fig.6a) with higher resolution. Next, we added a sphere mask at the transmembrane domain and part region of stalk domain with local refinement method for improving the resolution of TM domain. Fortunately, we got a low-resolution map which depict four TM helices (Revised Supplementary Fig.6b). Though observation of TM local refinement map, we found that the boundaries of the G protein TM domain is 34 to 66, but not 40 to 72 in our previous version. We have also modified text in revision in line 140-146: *“Employing single-particle cryo-electron microscopy (cryo-EM) technology, we successfully obtained a high-resolution 3D map of the extracellular domain of LayV-G protein at 2.8 Å resolution, revealing a distinct tetrameric architecture. (Fig. 3a, Supplementary Fig. 5-7, and Table S1). A sphere mask in Local refinement method of cryo-SPARC was applied to resolve the TM domain of the full-length LayV-G protein. As a result, we obtained a low-resolution TM map with four cylindrical TM helices (Fig. 3a and Supplementary Fig. 6b and 7b).”*

Revised Supplementary Fig.6a, b Cryo-EM data processing

Figures:

1 – *What are the special characters in the consensus line of Supplementary Figure 1? They are not defined in the legend.*

We thank this reviewer for pointing this out. The alignment was performed using MultAlin (<http://multalin.toulouse.inra.fr/multalin>). The special characters for consensus symbols in the consensus line in MultAlin software represent different amino acids: ! is anyone of IV, \$ is anyone of LM, % is anyone of FY and # is anyone of NDQEBZ. We have added this information in our Supplementary Fig. 1 legend “**Supplementary Figure 1 Sequence comparison of representative Henipaviruses attachment glycoproteins. The sequences including Langya virus (LayV-G, GenBank: UUV47206.1), Mojiang virus (MojV-G, GenBank: YP_009094095.1), Nipah virus (NiV-G, Genbank: NP_112027.1) and Hendra virus (Hev, GenBank:NP_047112.2) were aligned using MultAlin 5.4.1 and ESPript 3.0. The cysteine residues forming disulfide bonds are marked with green numbers. The special characters for consensus symbols in the consensus line in MultAlin software represent different amino acids: ! is anyone of IV, \$ is anyone of LM, % is anyone of FY and # is anyone of NDQEBZ.**”.

2 - *Supplementary Figure 3, the legend lacks most of the relevant information that it should contain. What is “purification by detergent”?*

We thank this reviewer for this insightful comment. It is Supplementary Figure 5 in the revised version, and we have rewritten the details of the legend: “**Supplementary Figure 5 Cryo-EM sample purification, micrograph and representative 2D classification. a** The last purification step by size exclusion chromatography (SEC) in the presence of GDN. The fractions within the red box were pooled for cryo-EM sample preparation. **b** A representative micrograph and 2D class average images are displayed.”

3 – *Line 184, Supplementary Figure 6c is called out but there is no Supplementary Figure 6c. Also, the figure legend makes little sense: “head binding cause two places of stalk bend”*

We thank this reviewer for this critical comment. The supplementary Fig. 6 is changed to supplementary Fig. 9 in the revised version and we have rewritten the supplementary figure legend “**Supplementary Figure 9 The interaction between head and 4HB of LayV-G cause the helixes of stalk bend and twist. a** Each head contact with two adjacent 4HB subunits. The head1 interact with the red area from HB1 and HB2, and the

head2 interact with the pink area from HB2 and HB3. b Head1 binding cause two places (yellow triangle and red triangle) of HB1 and HB2 bend and twist, respectively.” and the mistake about supplementary Fig. 9 has now been corrected in the revised manuscript.

Minor:

1 – The title – what is so “special” about this particular protein structure?

We thank this reviewer for this insightful comment. Here, we use “special” to the cryo-EM structure of homotetrameric attachment glycoprotein from langya henipavirus, which is distinctive from other members of paramyxoviruses families with known tetrameric structures (Revised supplementary Fig. 8). We have changed the title to “*The cryo-EM structure of homotetrameric attachment glycoprotein from langya henipavirus*” in our revised manuscript to make it more concise.

2 – The text could use major editing to improve syntax and grammar.

Point taken. We have revised the manuscript thoroughly.

Reviewer #2:

We appreciate the reviewer's acknowledgment of the significance of our study. Below please find our point-to-point responses:

Major comments:

1. In Figure 1a for biochemical characterization of the LayV-G protein, authors did not provide experimental controls for ephrinB2 binding either in the gel-filtration profile nor the SDS-PAGE analyses. Data in Figure 1 did not provide any further biochemical evidence that purified recombinant protein produced from LayV-G sequences is folded correctly or is a functional protein. To address this concern, including NiV-G and MojV-G binding profiles with ephrinB2/B3 as positive and negative controls in Figure 1 or supplementary Figure 2 could strengthen these data.

We would like to express our sincere gratitude for these valuable suggestions. One of the key enhancements we have incorporated into our study is a positive control (Revised Supplementary Fig. 3). Specifically, we incubated NiV-G with ephrin-B2/B3 and subjected it to gel-filtration. This modification has yielded compelling results: a substantial shift in the peak position of the protein, as well as the formation of a relatively stable complex

between NiV-G and ephrin B2/B3, as evidenced by SDS-PAGE analysis under reducing conditions. Furthermore, to reinforce our findings, we performed BLI analysis using recombinant ephrin B2/B3, which yielded similar results demonstrating the binding interaction between NiV-G and ephrin B2/B3 (Revised Fig.1c). This additional evidence corroborates the conclusion that the ephrin B2/B3 constructs obtained through recombinant expression possess the native conformation and functional characteristics required for binding with NiV-G.

Revised Supplementary Fig. 3 Biochemical characterization of the NiV-G binding to ephrin B2 and ephrin B3

Revised Fig.1c Bio-layer Interferometry (BLI) data of Binding affinity of ephrinB2

2. *It is not clear that the purified ephrinB2 or ephrinB3 used are stable or functional proteins that resemble functional surface expressed ephrin's. Since the gel-filtration approach does not address this concern, consider using ELISA or flow cytometry to determine LayV-G binding capacity to susceptible cells, or a functional cell-cell fusion assay.*

Point taken. As previously mentioned, we have complemented our study with additional biophysical methods, such as BLI, to further validate our findings. Both the gel-filtration approach and BLI experiments unequivocally demonstrate a robust interaction between ephrin B2/B3 and NiV-G (Fig. 1a-c). This compelling evidence not only reinforces our primary results but also serves as a confirmation that the ephrin B2/B3 constructs we obtained through recombinant expression possess the native conformation and functional capabilities necessary for effective binding. We also performed flow cytometry to determine LayV-G's binding capacity to susceptible cells. To make sure the natural structure and functional of ephrin B2 and ephrin B3, we constructed full length ephrin B2 and ephrin B3 stably expression CHO cell line and evaluated the amount of soluble HN-V-G binding to the ephrin B2 or B3 expressed on the surface of CHO cells through flow cytometry staining (data are shown in Revised Supplementary Fig. 4). The HeV-G and NiV-G bind more to the CHO-ephrin B2 or B3 cell comparing to CHO cell, while the overexpression of ephrin B2 or B3 on the cell surface did not significantly enhance the amount of LayV-G and MojV-G binding to CHO cell, suggesting the ephrin B2 or B3 possess no binding affinity to LayV-G and MojV-G.

Supplementary Figure 4 HNV G protein binding assay by flow cytometer.

3. The authors did not indicate in the legend of Figure 1 nor the Methods section how they prepared the samples for SDS-PAGE, and which conditions, e.g. reducing or non-reducing, they used. In addition, information regarding the method of choice for protein staining must be included. From the results of the SDS-PAGE image in Figure 1, it is unclear what type of putative monomeric, dimeric, or tetrameric protein species were collected from gel-filtration. The authors should include results from a Coomassie stain or results of Western Blot analysis under non-reducing conditions stained against expressed proteins.

We greatly appreciate this insightful comment and have made several updates. Firstly, we have provided a more detailed description of the SDS-PAGE sample preparation conditions within the revised manuscript, ensuring transparency in our methodology. Additionally, to further elucidate the structural aspects of our study, we conducted Coomassie Blue (Revised Supplementary Fig. 2a) and Western blot (WB) analyses (Revised Supplementary Fig. 2b) on the purified LayV-G fraction and ephrin B2 under non-reducing conditions. These analyses yielded noteworthy insights into the quaternary

structure of LayV-G and the stability of ephrin B2. Specifically, our findings indicate that LayV-G predominantly exists in the form of tetramers, a result consistent with our observations through cryo-EM data. Furthermore, the analysis of ephrin B2 reveals its stable monomeric conformation.

Revised Supplementary Fig. 2 Biochemical characterization of the LayV-G and ephrin B2 with SDS-PAGE and Western blot

4. The reported observation that LayV-G adopts a “heads-down” conformation is very interesting. Nevertheless, the authors should have explained or elaborated in the Discussion how this heads-down conformation compares to MojV and is related to possible interactions with the fusion protein and with receptors.

We thank this reviewer for pointing this out. In our work, we resolved the homotetrameric G protein 3D structure firstly, and there are neither full-length nor the ectodomain MojV-G have been reported so far. Therefore we now add the following sentences in Discussion part (line 230-251): *“Based on their close genetic relatedness and comparison to members of the HNV genus, both LayV and MojV could be classified as a new species within this viral group 5. The amino acid sequence comparison of LayV-G and MojV-G reveals an 86% sequence identity and 94% sequence similarity, while LayV-G shares only 19% identity and 64% similarity with NiV-G (Supplementary Fig.1). To date, no comprehensive reports on the full-length or ectodomain of LayV-G or MojV-G have been published, except for a crystal structure analysis of the MojV-G head 25. Based on this prior structural and functional research, it was observed that the head domain of MojV-G*

displays unique antigenic characteristics and employs a distinct mechanism for host-cell recognition when compared to other members of the HNV genus 25. Here, we investigated the antigenic divergence and receptor-binding properties of full-length LayV-G and MojV-G. We found that the several human mAbs, specifically target the HeV-G and NiV-G proteins³⁵, failed to bind to LayV-G and MojV-G (Fig. 2a, Supplementary Fig. 4). Polyclonal LayV-G antibody derived from rabbits show clear divisions appear among HeV-G and NiV-G proteins, although LayV-G mouse antisera exhibited minimal cross-reactivity with HeV-G (Fig. 2b). We also used two mAb of MojV-G to assess antigenicity of attachment glycoprotein of HNV, and found that one of MojV-G mAb are able to bind to LayV-G antigen with the desired activity (Fig. 2b). Interestingly, these findings suggest that LayV-G exhibits unique antigenic properties, differing from the G protein found in typical HNVs. It also appears to share a significant antigenic resemblance with MojV-G, possibly owing to similarities in both primary and quaternary structures.” Besides, we also add some speculation about the hypothetical working model in line 271-288: “The membrane fusion and entry process of most paramyxoviruses is reliant on a poorly understood activation step that involves receptor recognition, activation of the F-protein by the RBP, and subsequent membrane fusion. In a recent study conducted by Tara et al., the resting state RBP-F complex of human parainfluenza virus 3 was visualized on the surface of authentic viruses⁵⁰. This visualization provided insights into the mechanism by which the RBP head domains helps stabilize the pre-fusion state of the F-protein prior to receptor engagement. The stalk domain of RBP plays a crucial role in F activation and determines the specificity of the virus, while the rotation of the RBP heads may induce the movement of the stalk helices in relation to one another. Three intriguing questions for LayV-G possible interactions with the F protein and with receptors can be raised. (i) How does the resting state of LayV-G contribute to the stabilization of the metastable prefusion state of LayV-F? (ii) If the LayV-G receptor binding site is situated on the same side as other paramyxoviruses, what factors facilitate the transition from the “down” to “up” conformation of the head domains? (iii) How does the receptor engagement trigger the metastable pre-fusion LayV-F to undergo the series of structural transitions that result in fusion of the viral and cellular? Taken together, we have proposed a hypothetical working model for the infection of LayV based on previous studies.”

5. The authors confirmed that LayV-G does not bind ephrinB2 or ephrinB3. However, this finding had already been shown and that original reference (Zhang et al., NEJM 2022) should be cited. The following 3 primary references for important papers reporting ephrinB2 and ephrinB3 as HeV and NiV receptors are also missing and should be cited: a) Negrete OA, Levroney EL, Aguilar HC, Bertolotti-Ciarlet A, Nazarian R, Tajyar S, Lee B. ephrinB2 is the entry receptor for Nipah virus, an emergent deadly paramyxovirus. Nature. 2005 Jul 21;436(7049):401-5. doi: 10.1038/nature03838. Epub 2005 Jul 6. PMID:

16007075

b) Bonaparte MI, Dimitrov AS, Bossart KN, Crameri G, Mungall BA, Bishop KA, Choudhry V, Dimitrov DS, Wang LF, Eaton BT, Broder CC. *ephrin-B2 ligand is a functional receptor for Hendra virus and Nipah virus. Proc Natl Acad Sci U S A. 2005 Jul 26;102(30):10652-7. doi: 10.1073/pnas.0504887102. Epub 2005 Jul 5 ; PMID: PMC1169237.*

c) Negrete OA, Wolf MC, Aguilar HC, Enterlein S, Wang W, Mühlberger E, Su SV, Bertolotti-Ciarlet A, Flick R, Lee B. *Two key residues in ephrinB3 are critical for its use as an alternative receptor for Nipah virus. PLoS Pathog. 2006 Feb;2(2):e7. doi: 10.1371/journal.ppat.0020007. Epub 2006 Feb 10 ; PMID: PMC1361355.*

We thank this reviewer this suggestion. To date, we have read the original reference (Zhang et al., NEJM 2022, PMID: 35921459) carefully again and didn't find the result about LayV-G does not bind ephrinB2 or ephrinB3 in this article. We have already cited this reference as the original identification for the LayV. Besides, the rest three references have been cited in the revised manuscript.

Minor comments:

1. *In the Abstract, the authors claimed that they determined the full-length structure of LayV-G, although the cryo-EM map lacked densities for the intracellular domain, portion of the transmembrane domain, and the neck domain. In this case, it is more accurate to report the determination of the structure of the LayV-G ectodomain.*

We thank this reviewer for this insightful suggestion. We first improved the cryo-EM structure processing and got a new density map (Revised Fig. 3a) with higher resolution. Next, we add a sphere mask at the transmembrane domain and part region of stalk domain with local refinement method for improving the resolution of TM domain. Fortunately, we got a low-resolution map which depict four TM helices (Revised Fig. 3a and Supplementary Fig. 6,7). Though observation of TM local refinement map, we found that the boundaries of the G protein TM domain is 34 to 66 a.a, but not 40 to 72 a.a in our previous version. We have also modified text in result part (The overall architecture of the LayV-G homotetramer) in line 140-146: *“Employing single-particle cryo-electron microscopy (cryo-EM) technology, we successfully obtained a high-resolution 3D map of the extracellular domain of LayV-G protein at 2.8 Å resolution, revealing a distinct tetrameric architecture. (Fig. 3a, Supplementary Fig. 5-7, and Table S1). A sphere mask in Local refinement method of cryo-SPARC was applied to resolve the TM domain of the full-length LayV-G protein. As a result, we obtained a low-resolution TM map with four cylindrical TM helices (Fig. 3a and Supplementary Fig. 6b and 7b).”*

2. *In the Abstract incomplete sentence. "In this study, we conducted experiments to investigate LayV-G's binding capabilities." Binding capabilities to what? It would be clearer if they mention actual binding to the ephrin receptor in this sentence.*

We thank this reviewer for this good suggestion. In order to make the expression and the experimental data more rigorous, we set up Bio-Layer Interferometry (BLI) experiments to validate the binding property of LayV-G with known henipavirus receptors ephrin B2/B3 (Fig. 1c) and modified the text in manuscript in line 27-28 "*In this study, we conducted experiments to investigate LayV-G's binding capabilities to known henipavirus receptors.*"

3. *In Figure 1b the authors did not indicate the source or company name of anti-LayV-G polyclonal antibodies used for their ELISA analysis.*

Thank you for pointing it out. We have added the source and company information in the method part in the supplementary information and the results of ELISA have been replaced in the revised Fig. 2.

4. *Including MojV-G glycoprotein for screening with anti-HeV-G mAb can help to obtain a full assessment of LayV-G antigenicity. Consider using anti-MojV monoclonal or polyclonal antibodies to identify potential cross-reactivity with MojV-G to elaborate on the similarity between LayV-G and MojV-G.*

We thank the reviewer for the insightful comment. We included MojV-G monoclonal, LayV polyclonal antibody to identify potential cross-reactivity with MojV-G to elaborate on the similarity between LayV-G and MojV-G. The finding showed MojV-G and LayV-G shared similar antigenic properties in some extent. See in the revised Fig. 2b,c.

5. *In the Methods section Protein expression and purification, correct 2.0 x 10⁶ cells/mL to 2.0 x 10⁶ cells/mL.*

Point taken. We have corrected it.

6. *In the Methods section Cryo-EM sample preparation and data acquisition, add space after the period see "...GIF Quantum energy filter. Movie stacks..."*

Point taken. We have corrected it.

7. In the Methods section (Model building and structure refinement), correct *alphafold2* to *AlphaFold 2*.

Point taken. We have corrected it.

8. Line 177 remove space. Correct “in head 2” to “in head2”.

Point taken. We have corrected it.

9. The claimed novel structure shared between LayV-G and MojV-G, but absent in NiV and HeV, as described in paragraph at lines 151-159 “two antiparallel β strands from the N- and C-termini, which are further stabilized by disulfide bonds between Cys188 and Cys604 in blade 6 (Fig 3a)” should be better depicted and labeled in Fig. 3a so that any reader can easily identify this novel structure.

We thank this reviewer for pointing this out. Here, we change the Fig. 3 to the Fig. 4 now. The Revised Fig. 4 have highlighted the Cys188, Cys604 and Asn189 (N-linked glycosylation site).

Revised Figure 4 The feature of head domain of LayV-G and structure alignment with several HNVs.

10. Acknowledgments and author contributions section: It needs to be clarified why Dr. Wenjing Liu is not a coauthor on the paper. He performed initial cloning and protein

purifications, as S.W. and H.Y. Could the authors justify why the contributions of S.W. and H.Y. but not Wenjing Liu are worth authorship on the paper?

Point taken. Dr. Wenjing Liu used to be a postdoc in our laboratory who had done some initial sub-clone work and protein expression test of LayV-G and then left our lab soon. The significant work of cloning, purification, and biochemical assays are done by S.W. and H.Y. We acknowledged her initial work for this project but she didn't contribute significantly to the final result. So we don't think she deserve the authorship for this manuscript.

11. Line 289: Reference paper #7 Zhao, Y et al. is inappropriate. This publication describes a large-scale integrative analysis of psychiatric disorders and is irrelevant to the topic cited in the Introduction section (Line 51).

Point taken. The mistake may result from an automated update of endnote software. We have corrected it.

12. Line 358 and Line 371: In the References section, May A.J at el. J Virol (2023) was referenced twice as reference #34 and reference #40.

We thank this reviewer for the careful examination. We have corrected this mistake.

13. References for receptor-induced conformational changes in other henipaviral G proteins such as NiV or HeV should be added in the Discussion. Examples are Liu et al, 2013 and Liu et al., 2015.

We thank this reviewer for pointing this out and we have cited these works in the discussion section.

Reviewer #3:

We thank this reviewer for the critical and constructive comments, especially the detailed points. Below please find our point-to-point response:

Major Comments:

1) In figure 1 when describing the lack of cross-reactivity with known HeV antibodies, it would be helpful to add a figure that compares sequence alignments between LayV, MojV, HeV and NiV. Are there regions of conservation that could represent potential cross-reactive epitopes? Are the antigenic epitopes mapped for the antibodies described? Maybe consider structural conservation maps (i.e. consurf) and highlight antigenic epitopes for the HeV Abs tested. This sort of figure would provide stronger support the

final statement (line 113-114) in the paragraph. Was binding to MojV G evaluated?

We thank this reviewer for pointing this out. We have added the figure that compares sequence alignments between LayV, MojV, HeV and NiV (Revised Supplementary Fig. 1). There is no access to the information of the detailed epitopes in the reference of HeV antibodies (PMID:34469726), and no known structures of antibody-NiV-G/HeV-G complex. We have added the result of cross-reactivity to MojV G in the revised manuscript (Revised Fig. 2).

Supplementary Fig. 1 Sequence comparison of representative Henipaviruses attachment glycoproteins.

Revised Fig.2 The antigenicity of LayV-G protein in comparison to HeV-G, NiV-G and MojV-G proteins.

2) In Figure 3a, were there supposed to be zoom in figures (light grey boxes). If not, it would be helpful. In a supplementary figure, add overlays highlighting blade 6 and blade 4 between LayV/MojV and LayV/HeV, NiV or CedV.

We thank this reviewer for pointing this out. Here, we change the Fig. 3 to the Fig. 4 now. In addition, regarding the Fig. 4a, we are sorry for the mistake here that the tip of blade5 was named as blade6 tip. Now, we have modified Fig.4b to correct this error and optimized the cartoon style in the figure. The overlays highlighting blade5 tip and blade 4 tip between LayV/MojV and LayV/HeV, NiV or CedV were added in Fig4b now.

3) Line 194-196, please show through overlays or side by side images, the differences between LayV G and other HNVs.

We thank this reviewer for this critical comment. We have added a graph to the supplementary figures (Revised supplementary Fig. 8) showing the comparative structural gallery of known nearly extracellular domain of paramyxovirus attachment

glycoprotein architectures and our Lay-G structure. The figure can better illustrate the differences of heads positions of attachment glycoprotein in the different paramyxovirus.

4) Line 203-205 – There is no strong evidence that this is the case. This also goes to the statement in the abstract mentioning “the F protein influencing membrane fusion by obstructing the charged regions of the stalk”. I would exclude “by obstructing the charged regions of the stalk”. This is a hypothesis which can be discussed in the discussion but no firm conclusions can be made at this time.

Thank you for your advice. We have deleted “by obstructing the charged regions of the stalk” in the abstract.

5) Line 233-241 – the model proposed in Figure 6 is the same as that proposed in May et al 2023. The difference is in highlighting the possible role of the electrostatic 4HB domain. I would reference May et al. when describing the proposed model.

Point taken. We have cited related references about putative working model for LayV infection here including May et al 2023.

Minor Comments

1) A better description of the HNV attachment protein, in its many forms (HN/H/G) may be receptor binding protein (RBP).

We thank the reviewer for this good suggestion. Now, we have carefully followed the suggestion in line 62 (“attachment glycoprotein” to “RBP”) and line 64 (“HN/H/G proteins” to “RBP”).

2) Line 64, need a space at end of sentence.

Point taken. We have corrected it.

3) Line 111, anti-LayV-G polyclonal antibodies should be antibody (it is a single sample).

Point taken. We have corrected it in Line 128 “the commercial anti-LayV-G polyclonal antibodies” to “a commercial anti-LayV-G polyclonal antibody”.

4) Line 136-7, add that the N-glycosylation site at Asn 189 is unique to LayV. Was this commented on in the discussion section?

We thank the reviewer for this good suggestion. We add this context in the sentence in line 154-156: *“Moreover, an N-glycosylation site is present at amino acid Asn189 on the head domain of each protomer, which is unique to LayV (Fig. 3b-d).”*. We also added a section on glycosylation in the Discussion (line 253-265): *“Moreover, two independent research groups reported the structures of LayV-F in both the prefusion and post-fusion states, which provided crucial insights into the activation of LayV-F^{36,41}. One of these studies has demonstrated that LayV-F and MojV-F exhibit distinct glycosylation modifications and antigenic profiles, despite their structural similarity to NiV⁴¹. According to the results reported by Ilona et al.²⁵, the ectodomain of MojV-G contains only four potential N-linked glycosylation sites, while the β -propeller domain of MojV-G exhibits a distinct feature of low glycolysis compared to HNV-RBPs^{21,28,42-45}. In our near full-length structure of LayV-G, only a single glycosylation site (Asn189) is observed. Due to the absence of full-length MojV-G structure data, it remains uncertain whether the glycosylation profiles of MojV-G are consistent with LayV-G. Therefore, a thorough investigation is required to ascertain the impact of glycosylation on the virulence and antigenic cross-reactivity of LayV-G.”*

5) Figure 1a – was there a positive control demonstrating binding to ephrin B2 with either NiV G or HeV G?

We thank the reviewer for this good suggestion. We have added the positive control about the SEC co-migration assay in the revised manuscript. We demonstrated that ephrin B2/ephrin B3 and NiV-G co-migrate in a protein gel (Supplementary Fig. 3).

Revised Supplementary Fig. 3 Biochemical characterization of the Niv-G binding to ephrin B2 and ephrin B3

6) *In figure 2 – the use of lighter shades, makes it easier to see the cryo-EM structure.*

Point taken. We have corrected the related figure.

7) *In figure 3b – use of lighter shades would make it easier to see structural divergence and/or overlap.*

Point taken. We have corrected the related figure.

8) *Please orient figure 3a and b (top) in the same direction.*

Point taken. We have changed it.

9) *What are blades? I didn't see a clear definition.*

Accepted, we have added the definition of 'blade' in revised manuscript (line 165-167: *"The head domains of LayV-G (residues 187 to 607 a.a) demonstrate a globular six-bladed β propeller fold, wherein each blade consists of four antiparallel β -strands that are interconnected by seven internal disulfide bonds"*).

10) *In Figure 4a, label the red boxes in the corner to correspond to b,c,d as appropriate. Again, lighter shades, especially in b, c and d would make it easier to see what the authors are trying to show.*

Accepted, we have redrawn the Fig. 4 and it is Fig. 5 now. We also corrected the box color correspond to b,c,d as appropriate.

11) *Figure 5 legend, d electrostatic surface...should be e electrostatic surface...*

Point taken. We have corrected it.

12) *In Figure 5, mark positive (blue) and negative (red) charge.*

Accepted, we have added the positive and negative bar and rewritten the legend of Fig. 5 and its Fig. 6 now.

13) *Lines 162-169 – paramyxoviral should be paramyxovirus (3 instances)*

We thank the reviewer for pointing out this mistake. We have corrected accordingly.

14) *Line 175, need a space between head1.*

Thank your advise. Consider this is a name of a domain, we decided to unify the different head domain's name with no space between head and its number. We have now fixed these names in our revised manuscript.

15) *Reference 34 and 39 are the same.*

We thank this reviewer for the careful examination. We have corrected it.

16) *Supplementary figure 1 legend Lay-G should be LayV-G (correct twice), also consider selected different colors than red and green for those with color blindness.*

Accepted. We have corrected it.

17) *Supplementary Figure 2 legend, in vitro should be italicized.*

Accepted. We have corrected it.

18) *Supplementary Figure 3 legend, full-lengh should be full-length.*

Point taken. We have corrected it in the revised version.

19) *Supplementary Figure 3 – please include a bar of length (nm) for the EM micrographs.*

Point taken. We have corrected it in the revised version.

20) *Supplementary Figure 6 legend – in vitro should be italicized, headings a and b need to be switched.*

Point taken. We have corrected it in the revised version.

21) *The HeV antibodies used should be described in the methods and reporting summary.*

Point taken. We have added it into the revised version.

We sincerely appreciate all three reviewers' input again.

REVIEWERS' COMMENTS

Reviewer #1 (Remarks to the Author):

all comments have been properly addressed

Reviewer #2 (Remarks to the Author):

My prior critiques have been addressed well. My only new suggestion is that if the LayV G sequences was at all changed from the original published LayV G sequence, the new sequences should be shown or deposited as a new sequence submission and referenced accordingly.

Reviewer #3 (Remarks to the Author):

Dear Authors,

Thank you for so thoroughly addressing the comments received from all three reviewers. The revised manuscript is much stronger with the additional data and analyses.

A few comments

- 1) Please check for consistency of Ephrin B2 vs EphrinB2 throughout.
- 2) in methods and reporting summary - what version of Prism is used?
- 3) in methods, line 117-126 - how were samples prepared? +/- reducing agent, +/- heat, etc.? What is block solutions and what are antibodies diluted in?
- 4) in supplementary fig 2 - is there a possible explanation for why the monomeric band between a and b looks to be a different MW? a=lower than 100kD, b=almost exactly 100kD
- 5) abstract line 32 "various"
- 6) in methods line 41 - LayV

7) in methods line 50-56 - HEPES, Tween-20, His, check spacing around ().

8) in methods line 58-68, 97-111 - 1xPBS, 1xPBST (define once what percentage Tween-20 is in buffer)

9) in methods line 159 - "Coot"

10) Supp fig 9 - "helices"

11) in main manuscript - line 127 replace "as well as" with "or", line 131 replace "Both" with "Neither", remove "two", line 132 replace "did not bind" with "bound", line 250 need "." at end of sentence, line 273 italicize "et al."

Thank you.

Response to Reviewers:

We appreciate the comments from reviewers. Below please find our point-to-point responses.

Reviewer #1:

all comments have been properly addressed.

We sincerely appreciate this reviewer's input.

Reviewer #2:

My prior critiques have been addressed well. My only new suggestion is that if the LayV G sequences was at all changed from the original published LayV G sequence, the new sequences should be shown or deposited as a new sequence submission and referenced accordingly.

We thank this reviewer for the insightful suggestion. The protein sequence for recombinant expression is consistent with the original published LayV-G sequence.

Reviewer #3:

Dear Authors,

Thank you for so thoroughly addressing the comments received from all three reviewers. The revised manuscript is much stronger with the additional data and analyses.

A few comments:

1) Please check for consistency of Ephrin B2 vs EphrinB2 throughout.

Point taken. We have carefully replace each Ephrin B2 into EphrinB2 in the revised manuscript and supplementary information file.

2) in methods and reporting summary - what version of Prism is used?

Point taken. GraphPad Prism 8.0 was used in the research. We have revised it.

3) in methods, line 117-126 - how were samples prepared? +/- reducing

agent, +/- heat, etc.? What is block solutions and what are antibodies diluted in?

Corrected. The information of samples preparing conditions and the solutions what antibodies diluted in already added in methods. All samples are not heated. And we have modified the text in manuscript in line 475-478 "*The samples were prepared using 5xSDS-PAGE Loading Buffer containing 0.25 M Tris-HCl (pH 6.8); 10% SDS; 0.5% BPB; 50% glycerol; 5% (W/V), with or without 100 mM dichlorodiphenyltrichloroethane (DTT) depending on reduced or nonreduced conditions analyses.*"

4) in supplementary fig 2 - is there a possible explanation for why the monomeric band between a and b looks to be a different MW? a=lower than 100kD, b=almost exactly 100kD

We thank this reviewer for the insightful comment. We have noticed that MW behaves slightly differently in SDS-PAGE vs. WB. It can be noticed that in WB, whether it is a tetramer, dimer or monomer band, there is an upward shift relative to the corresponding position in SDS-PAGE. We proposed that this might be due to some deformation of PVDF during the WB transfer process. This situation is caused by the difference in SDS-PAGE gel or the tilt during scanning. In order to address this concern, we have re-performed SDS-PAGE and WB, and used the same PAGE gel for Coomassie Brilliant Blue staining and WB. In this result, the positions of the corresponding bands in SDS-PAGE and WB are consistent, so we think it should be the difference between SDS-PAGE gels and the smeariness of the LayV-G band when the gel running time is too long. We have revised the supplemental figure 2.

5) abstract line 32 "various"

Corrected.

6) in methods line 41 – LayV

Point taken. We have corrected it.

7) in methods line 50-56 - HEPES, Tween-20, His, check spacing around ().

Thank you for pointing it out. We have corrected it.

8) in methods line 58-68, 97-111 - 1xPBS, 1xPBST (define once what percentage Tween-20 is in buffer)

Corrected. The information of what percentage Tween-20 is in buffer has been added in in manuscript in line 419 *“(containing 0.1% Tween-20 in PBS)”*

9) *in methods line 159 - “Coot”*

Point taken.

10) *Supp fig 9 - “helices”*

Point taken. We have corrected it

11) *in main manuscript - line 127 replace “as well as” with “or”, line 131 replace “Both” with “Neither”, remove “two”, line 132 replace “did not bind” with “bound”, line 250 need “.” at end of sentence, line 273 italicize “et al.”*

Point taken. We have corrected it.

We thank these reviewers again for their kind patience.